# Trustworthy and Responsible AI for Human-Centric Autonomous Decision-Making Systems

**Farzaneh Dehghani**                                        *farzaneh.dehghani@ucalgary.ca*
*Department of Biomedical Engineering, University of Calgary*

**Mahsa Dibaji**                                        *seyedemahsa.dibaji@ucalgary.ca*
*Department of Electrical and Software Engineering, University of Calgary*

**Fahim Anzum**                                        *fahim.anzum@ucalgary.ca*
*Department of Computer Science, University of Calgary*

**Lily Dey**                                        *lily.dey@ucalgary.ca*
*Department of Computer Science, University of Calgary*

**Alican Basdemir**                                        *alican.basdemir@ucalgary.ca*
*Department of Philosophy, University of Calgary*

**Sayeh Bayat**                                        *sayeh.bayat@ucalgary.ca*
*Department of Geomatics Engineering, University of Calgary*

**Jean-Christophe Boucher**                                        *jc.boucher@ucalgary.ca*
*Department of Political Science, University of Calgary*

**Steve Drew**                                        *steve.drew@ucalgary.ca*
*Department of Electrical and Software Engineering, University of Calgary*

**Sarah Elaine Eaton**                                        *seaton@ucalgary.ca*
*Werklund School of Education, University of Calgary*

**Richard Frayne**                                        *rfrayne@ucalgary.ca*
*Department of Radiology, University of Calgary*

**Gouri Ginde**                                        *gouri.deshpande@ucalgary.ca*
*Department of Electrical and Software Engineering, University of Calgary*

**Ashley Harris**                                        *ashley.harris2@ucalgary.ca*
*Department of Radiology, University of Calgary*

**Yani Ioannou**                                        *yani.ioannou@ucalgary.ca*
*Department of Electrical and Software Engineering, University of Calgary*

**Catherine Lebel**                                        *clebel@ucalgary.ca*
*Department of Radiology, University of Calgary*

**John Lysack**                                        *john.lysack@ucalgary.ca*
*Department of Radiology, University of Calgary*

**Leslie Salgado Arzuaga**                                        *leslie.salgadoarzuag@ucalgary.ca*
*Department of Communication, Media, and Film, University of Calgary*

**Emma Stanley**                                        *emma.stanley@ucalgary.ca*
*Department of Biomedical Engineering, University of Calgary*

**Reviewed on OpenReview:** *https://openreview.net/forum?id=1k833OTHpI*

**Roberto Souza**                                                     roberto.souza2@ucalgary.ca
*Department of Electrical and Software Engineering, University of Calgary*

**Ronnie de Souza Santos**                                      ronnie.desouzasantos@ucalgary.ca
*Department of Electrical and Software Engineering, University of Calgary*

**Lana Wells**                                                               lmwells@ucalgary.ca
*Faculty of Social Work, University of Calgary*

**Tyler Williamson**                                               tyler.williamson@ucalgary.ca
*Centre for Health Informatics, University of Calgary*

**Matthias Wilms**                                                   matthias.wilms@ucalgary.ca
*Department of Radiology, University of Calgary*

**Mark Ungrin**                                                            mdungrin@ucalgary.ca
*Faculty of Veterinary Medicine, University of Calgary*

**Marina Gavrilova**                                                       mgavrilo@ucalgary.ca
*Department of Computer Science, University of Calgary*

**Mariana Bento**                                          mariana.pinheirobent@ucalgary.ca
*Department of Biomedical Engineering, University of Calgary*

## Abstract

Artificial Intelligence (AI) has paved the way for revolutionary decision-making processes, which, if harnessed appropriately, can contribute to advancements in various sectors, from healthcare to economics. However, its black box nature presents significant ethical challenges related to bias and transparency. AI applications are hugely impacted by biases, presenting inconsistent and unreliable findings, leading to significant costs and consequences, highlighting and perpetuating inequalities and unequal access to resources. Hence, developing safe, reliable, ethical, and Trustworthy AI systems is essential. Our interdisciplinary team of researchers focuses on Trustworthy and Responsible AI, including fairness, bias mitigation, reproducibility, generalization, interpretability, explainability, and authenticity. In this paper, we review and discuss the intricacies of AI biases, definitions, methods of detection and mitigation, and metrics for evaluating bias. We also discuss open challenges with regard to the trustworthiness and widespread application of AI across diverse domains of human-centric decision making, as well as guidelines to foster Responsible and Trustworthy AI models.

## 1 Introduction

Artificial Intelligence (AI) represents the frontier of computer science, enabling machines to emulate human intelligence and perform tasks that were once exclusive to human capabilities (Briganti & Le Moine, 2020). This rapid progression in AI, driven by Machine Learning (ML) and Deep Learning (DL) innovations, has catalyzed breakthroughs across various industries, including business, communication, healthcare, and education, among others. Utilizing state-of-the-art computational resources, the AI models are trained on extensive datasets and can be used for decision-making on unseen data. Recent advancements in AI algorithms and feature engineering techniques have played a pivotal role in transforming various human-centric fields, notably, healthcare (Esteva et al., 2019), image and text generation (Epstein et al., 2023), biometrics and cybersecurity (Gavrilova et al., 2022), online social media opinion mining (Anzum & Gavrilova, 2023), autonomous driving vehicles (Ma et al., 2020), and beyond.

Despite the impressive capabilities exhibited by recent AI-based systems, a significant challenge lies in their inherent black box nature. Due to the lack of explainability and interpretability of AI models, establishing trust among end users has become critical (von Eschenbach, 2021). Therefore, to ensure trustworthiness in AI-empowered systems, it is imperative not only to improve the model's accuracy but also to incorporate explainability and interpretability into the model's architecture and decision-making process. Interpretability refers to the ability to explain or provide meaning in a way humans can understand, while explainability

involves providing details or reasons to facilitate or clarify understanding of AI systems (Arrieta et al., 2020a). Human-Centered Explainable AI (HCXAI) transcends purely technical interpretability by generating context-aware, interactive explanations tailored to users' cognitive processes and decision-making requirements. Research shows that explainability can ensure trustworthiness in AI-based models' decisions by visualizing the factors affecting the result, leading to fair and ethical analysis of the model (Schoenherr et al., 2023). Foundation models like large language models (LLMs) exhibit complex emergent behaviors that challenge traditional interpretability methods, prompting new approaches like mechanistic interpretability to analyze their internal circuits. These technical advances must integrate human-centered design principles to ensure explanations are meaningful, context-aware, and actionable across diverse stakeholders.

Although interpretability and explainability are essential to foster trust in AI systems, they are not sufficient alone. Beyond transparency, ensuring fairness, accountability, and robustness is equally critical. AI algorithms have raised issues related to bias, fairness, privacy, and safety, especially when human data is being used, avoiding perpetuating stereotypes and reinforcing inequalities (Anzum et al., 2022). Therefore, to develop an AI system that can be trusted by its end users and the communities it impacts, a holistic approach, combining technical explainability with ethical governance and stakeholder engagement, is necessary.

Preserving privacy when using human data is one of the most significant challenges in Trustworthy AI (Stahl & Wright, 2018). This can convince users that a model respects the privacy of data owners and does not reveal any information related to data. As for safety, not all AI-empowered systems with high performance can be considered safe, especially for human-subject involvement (Rasheed et al., 2022).

Here, the key research question is "what are the core concepts, challenges, and guidelines that underlie the principles of Trustworthy and Responsible AI within academic and industry contexts?". To answer this question, our paper makes the following contributions:

1. Determining and discussing several foundational principles in the landscape of AI including Trustworthy AI, Responsible AI, and Fairness in AI.

2. Exploring various principles regarding AI Governance, Regulatory Compliance, reproducibility, reliability, and communication.

3. Presenting a comprehensive overview of bias in terms of source and types, various techniques to determine and mitigate bias, and fairness evaluation metrics and frameworks.

4. Identifying the potential research gaps, challenges, and opportunities in the context of Trustworthy and Responsible AI by synthesizing and standardizing data from various applied human-centric domains, and providing our perspective on this context as a transdisciplinary team of researchers.

5. Discussing open challenges and limitations in the application of AI across diverse domains and proposing comprehensive guidelines to develop Trustworthy, fair, and reliable AI models.

## 2 Trustworthy and Responsible AI Definition

Despite extensive research on Trustworthy AI, there is a lack in definition and standardization, particularly across various disciplines where AI is applied. Therefore, our team, consisting of researchers from diverse fields such as communication, media and film, philosophy, social-behavioral sciences, public policy, political science, foreign policy, ethics, education, radiology, pediatrics, health informatics, veterinary medicine, biomedical research, computer science, and electrical and software engineering, provides a comprehensive perspective on Trustworthy and Responsible AI.

In the landscape of intelligent human-centric systems, several foundational principles stand out as guides for research and development. Prominent among these is Trustworthy AI. Trustworthy decision-making is defined as the ability of an intelligent computer system to perform a real-time task repeatedly, reliably, and dependably in complex real-world conditions (Lyu et al., 2021). This principle emphasizes that AI should be perceived as reliable by all its users, from individual consumers to entire organizations and broader society. Trustworthiness in AI is related to compliance with laws or robust system performance, also ensuring that AI adheres ethical guidelines (Díaz-Rodríguez et al., 2023). A key component of trustworthiness is transparency (Li et al., 2023). The decisions made by AI systems, the utilized data, and the processes governing them should be clear and interpretable to all stakeholders involved. Such clarity ensures that AI does not remain

an enigmatic "black box" and becomes an entity whose actions and decisions are understandable and, more importantly, accountable.

Fundamental principles of Trustworthy AI include beneficence, non-maleficence, autonomy, justice, and explicability. Beneficence in AI refers to designing and deploying systems that actively promote human and environmental well-being while respecting fundamental rights, with interpretations varying across frameworks, some focus narrowly on human welfare, while others extend to environmental and sentient beings' welfare or emphasize societal harmony. This principle aligns with trusting beliefs of benevolence and helpfulness, requiring AI to act in users' best interests without manipulation (Thiebes et al., 2021).

The principle of non-maleficence mandates that AI systems must be designed and implemented to avoid causing harm to both human populations and the natural environment. While closely related to beneficence (which focuses on actively promoting well-being), non-maleficence constitutes a distinct ethical obligation that is universally recognized across major AI governance frameworks. This principle manifests primarily through three critical dimensions: privacy protection, system security, and operational safety (Thiebes et al., 2021). Furthermore, non-maleficence extends beyond human impacts to encompass environmental protection, including mitigation of excessive computational energy demands, natural resource depletion, and carbon emissions from AI infrastructure (Vinuesa et al., 2020). From a trust perspective, this principle correlates strongly with system integrity, operational reliability, and procedural transparency, essential characteristics that ensure AI behaves predictably and adheres to established ethical parameters (Thiebes et al., 2021).

Autonomy in Trustworthy AI emphasizes maintaining a balanced relationship between human oversight and AI system independence, ensuring that humans retain meaningful control and the ability to intervene in decision-making processes. Different frameworks interpret this principle in varying ways, some focus on promoting human agency and oversight, while others highlight the need to restrict AI autonomy when appropriate. Autonomy serves as a safeguard to uphold system integrity and reliability and is conceptually linked to openness, which fosters trust. In the research domain, the concept spans robotics, human-agent interaction, and adjustable autonomy. Practically, organizations are encouraged to implement oversight mechanisms, such as keeping humans in the loop, to ensure that AI systems align with this principle (Thiebes et al., 2021).

Justice, often referred to as fairness, is a core ethical principle across all major Trustworthy AI frameworks. It emphasizes not legal compliance, but the ethical imperative to address historical inequities, ensure equitable distribution of AI benefits, and prevent the creation of new harms or biases. These goals are reflected in principles such as equity and anti-discrimination measures. Justice aligns with trust-related values like integrity and reliability, ensuring AI systems operate ethically and inclusively. In research, justice spans detecting and mitigating bias, particularly racial and systemic, in AI applications, with a strong focus in fields like healthcare and electronic marketplaces (Thiebes et al., 2021).

Explainability is a foundational principle of Trustworthy AI that encompasses both epistemological and ethical dimensions, promoting interpretability for understanding AI decisions and accountability for responsible use. While all major frameworks acknowledge this principle, they express it differently: some emphasize transparency, others intelligibility or interpretability. It supports trust by demonstrating competence and performance, enabling users to comprehend how AI systems work and ensuring accountability for their actions. As modern AI systems are often opaque black boxes, explicability is crucial for gaining user trust, meeting regulatory demands, and fostering responsible deployment. Research in this area focuses on developing inherently interpretable models (e.g., decision trees), post-hoc explanation methods (e.g., heatmaps), uncertainty quantification, and auditability, making explicability not only a technical challenge but a key enabler of broader ethical principles such as beneficence, autonomy, and justice (Thiebes et al., 2021).

Complementary to Trustworthy AI is the notion of Responsible AI. While these concepts overlap, they differ in emphasis. While Trustworthy AI defines what constitutes ethical and reliable AI, Responsible AI outlines how to build, govern, and deploy such systems in practice. Trustworthy AI focuses on the desired properties of AI systems, such as robustness, fairness, and explicability, to ensure ethical alignment and reliability (Cannarsa, 2021). In contrast, Responsible AI emphasizes the processes and practices required to achieve these properties, adopting a more tactical and governance-oriented approach (Arrieta et al., 2020a; Dignum, 2023). Responsible AI explicitly recognizes AI's socio-technical character, framing issues like bias, opacity, and unforeseen consequences not just as technical flaws but as systemic risks that demand ethical oversight and organizational accountability (Raji et al., 2020). It operationalizes trustworthiness

by promoting proactive harm reduction strategies (e.g., bias mitigation, environmental impact assessment), participatory design, and continuous auditing (Cheng et al., 2021).

Another significant notion in the AI landscape is Explainable AI (XAI). Fundamentally, from a computer science perspective, explainability is primarily about AI model transparency and comprehensible decision-making (Confalonieri et al., 2021). However, this technical view is expanded from the broader academic community, especially social science scholars. They advocate for a broader conception of explainability, one that goes beyond mere technical intricacies (Wang et al., 2019). This broader viewpoint underscores the importance of post-hoc interpretable explanations, which provide clarity on AI decisions (Gianchandani et al., 2023). It also highlights the need for including diverse stakeholders, from AI specialists to potential end-users, right from AI's developmental stages (Clement et al., 2023). This inclusive approach aims to make AI accessible, understandable, and trusted resource for all. To ensure model explainability and transparency, self-explainable models, being able to visually explain their decisions via attribution maps and counterfactuals, can be developed (Wilms et al., 2022). However, Explainable AI models do not release details on the decision-making process of models, which makes them unreliable and misleading. Thus, developing inherently interpretable models that are able to provide their own explanations is paramount (Rudin, 2019).

To clarify these concepts, interpretability refers to the extent to which a model's internal mechanics can be understood by humans. An interpretable model is transparent by design, like decision trees or linear models, where the process through which the input features combine to produce outputs is clear. Explainability on the other hand, focuses on providing understandable reasons for individual model predictions, especially for complex or "black box" systems. Techniques like saliency maps offer post-hoc explanations that clarify why the model made a certain decision (Arrieta et al., 2020b). Transparency is the overarching principle that encompasses both interpretability and explainability. It involves openness in the model's design, data, decision logic, and lifecycle. It ensures stakeholders can understand how and why a system works (Li et al., 2023).

Although Explainable AI contributes to user trust, human–machine collaboration, and algorithmic fairness, traditional XAI methods often fail in real-world applications because they ignore human factors. Indeed, different AI stakeholders—including end users, domain experts, AI developers/data scientists, regulators/auditors, and business decision-makers—have varying demands for explainability based on their goals, backgrounds, and decision scenarios. Therefore, there is a need to lead research and application in a more "human-centered" manner by integrating human factors into explanations. End users need simple, actionable, and non-technical explanations to understand AI decisions that affect them. These explanations should build trust, avoid jargon, and, where legally required, suggest remedial actions. Professionals need detailed, domain-specific explanations aligned with their expertise. The explanations must integrate seamlessly into their workflow, providing evidence-based rationales to support their judgment. AI developers and data scientists require technical, model-introspective explanations for debugging and improvement. They rely on XAI tools to identify feature importance, detect bias, or troubleshoot misclassifications. Regulators and auditors prioritize fairness, compliance, and accountability. They require explanations that reveal bias or regulatory violations. Business decision-makers need high-level, strategic insights to evaluate AI's operational impact, including cost–benefit analyses, risk summaries, and plain-language takeaways to guide policy or investment. These diverse demands highlight the importance of interactive explanations, where users can ask follow-up questions, and adaptive explanations that adjust their level of detail based on user feedback (Kong et al., 2024).

The rise of foundation models—particularly LLMs—has fundamentally reshaped the landscape of interpretability. These models exhibit emergent behaviors such as in-centext learning, reasoning, and abstraction, which arise without feature instruction (Berti et al., 2025). However, the traditional explainability tools such as saliency maps and feature attribution fall short in uncovering how such models internally organize knowledge or support generalization. Thus, researchers are increasingly turning to probing techniques that inspect the internal representation of LLMs across different layers. Li et al. (Li et al., 2024) introduced a probing framework grounded in reinforcement learning concepts to examine whether LLMs form state abstractions to support decision-making and generalization. Their findings show that certain model layers encode structured representations that resemble abstract states, suggesting that LLMs implicitly construct mental models of the world during training. Following the probing-based insights, a more structural interpretability approach termed mechanistic interpretability has emerged. This framework aims

to reverse-engineer the internal wiring of transformer models (e.g., GPT, LLaMA) to uncover how specific sub-networks—known as circuits—implement semantic or algorithmic functions. For instance, recent work by Lan et al. (Lan et al., 2024) demonstrated that sequence-continuation behaviors (e.g., predicting numeric or linguistic progressions) are mediated by shared circuit subgraphs in GPT-2 Small and Llama-2 7B, capturing algorithmic patterns across model variants and input types.

Importantly, advances in interpretability must be rooted in human-centered design principles to ensure they are actionable and meaningful across diverse user groups, including clinicians, regulators, developers, and the public. The field of HCXAI emphasizes that explanations must be more than technically accurate—they must be context-sensitive, interactive, and aligned with the cognitive and decision-making needs of end-users (Kong et al., 2024). Contrastive and part-based explanations, aligning with intuitive human reasoning processes, can improve users' ability to calibrate trust, provide feedback, and engage with AI systems more effectively (Kim et al., 2023). Furthermore, explainability is reframed as a socially situated process that involves transparency, participatory design, stakeholder inclusion, and institutional accountability. Such explanations should support real-world governance and oversight, adapting to users' goals, expertise levels, and organizational constraints (Ehsan et al., 2021). This human-centered perspective expands the function of XAI beyond individual interpretability, embedding it within broader sociotechnical systems to enhance trust, usability, and ethical deployment.

Lastly, Fairness in AI is a principle that acts as a safeguard against discriminatory practices (Madaio et al., 2022). An AI system, regardless of its complexities or application, should be devoid of unfair biases. It should not favor or marginalize any group based on social, demographic, or behavioral attributes. Thus, fairness is not just passively avoiding discrimination; it is an active commitment. Even if input data carries biases, the AI system's outputs should remain impartial, ensuring equitable algorithmic decisions for all (Dibaji et al., 2023). An experience developed by Fujitso in japan showed the challenge of assessing fairness in different contexts and scenarios, due to the cultural and societal differences. Researchers demonstrated the benefits of embracing socio-technical solutions to mitigate bias in the system outcomes (Research & Development, 2023).

While fairness and justice are related principles in trustworthy AI, they operate at different levels and require different approaches. Fairness in AI refers to the equitable treatment of individuals and groups by algorithmic systems. It focuses on ensuring that AI systems do not produce discriminatory outcomes based on protected characteristics such as race, gender, age, etc. Justice in AI encompasses broader considerations beyond statistical fairness, incorporating moral, legal, and societal dimensions. AI practitioners must address both to create systems that are not only technically sound but also socially responsible (Naudts & Vedder, 2025).

One growing concern surrounding AI development is its environmental impact. While normative approaches to AI ethics have largely focused on issues such as transparency, privacy, safety, responsibility, bias, and discrimination, the environmental consequences of AI should not be overlooked. The training of large-scale AI models requires substantial electricity, which contributes to greenhouse gas emissions. Moreover, significant volumes of water are needed to cool the data centers that power these systems. Beyond resource consumption, AI technologies can negatively affect ecosystems and wildlife. For instance, autonomous driving systems may disturb animal habitats, alter migration patterns, or even pose direct threats to certain species—consequences that may ultimately affect human populations as well. Therefore, environmental considerations must be an integral part of any ethical evaluation of AI systems (van Uffelen et al., 2024).

Another key concern relates to personal privacy (Stahl & Wright, 2018). If not properly regulated and without proper consent, AI may pose threats to personal freedom and human rights. Moreover, given AI's ability to process massive data volumes, preserving and respecting the richness of diverse cultures becomes imperative.

The advent of generative AI (Muller et al., 2022) introduces a new set of challenges. Such AI models, trained on extensive online data sets, can produce content that can blur the lines between reality and fiction (Jo, 2023), exacerbating disinformation. The old digital belief, "seeing is believing", is rapidly being rewritten. Companies, researchers, and developers have a collective responsibility to implement stringent safeguards, transparently disclose the origins of AI-generated content, ensuring that the digital ecosystem remains a source of reliable information.

Shifting focus into specific domains, healthcare presents its own set of unique challenges. Preserving patient data sanctity is paramount by removing individual patient, known as anonymization. While AI models in

healthcare revolutionize diagnostics, treatments, and patient care, they require vast amounts of diverse data for effective training (Tom et al., 2020). However, the sharing of such data between institutions often hits a wall due to stringent privacy concerns. Federated learning (FL), where data remains in its original location, is a possible solution to this challenge.

It is worth mentioning that, despite the effectiveness of anonymization and FL, these methods remain vulnerable to various privacy and security threats. Anonymized datasets remain vulnerable to re-identification attacks, especially when linked with external public or semi-public data. Also, anonymization often removes or distorts valuable information, which can impair the performance of AI models. Additionally, the process is computationally demanding, requiring resource-intensive pipelines to identify, transform, and verify sensitive data (PATCHIPALA, 2023). Federated Learning, while promising for preserving data privacy, remains susceptible to significant security and privacy threats. A major vulnerability lies in adversarial attacks, such as model poisoning, where malicious participants manipulate local data or updates to degrade model performance. Despite not sharing raw data, FL can still leak sensitive information through exchanged model updates, which attackers can use to infer private attributes, reconstruct training data, or identify user membership. Additionally, the central server, a key component in coordinating FL, poses a security risk if compromised or monitored. Similarly, malicious clients can exploit FL's decentralized design to inject harmful data with limited oversight (Zhu et al., 2025).

These challenges require the AI deployment respecting the individual, society, and the environment. Such an approach ensures that as we advance into the future, we proceed with caution, responsibility, and respect for ethical considerations.

## 3 Governance for Human-Centric Intelligence Systems

AI Governance emerges as a multifaceted discipline, crucial for ensuring that AI systems operate within a well-defined and ethical framework during the whole AI-life cycle. Governance is not just about overseeing AI models; it extends to the data use for model development. Effective data governance mandates that data acquisition, storage, and utilization presents ethical and integrity standards (Janssen et al., 2020). As AI systems grow in complexity and reach, mechanisms like federated learning gain prominence, advocating for data sharing without sacrificing privacy. This approach encompasses self-regulatory practices, including ethics and impact assessments at early design stages, and is complemented by practices that focus on data reuse and ethics, establishing guidelines on data re-use. However, FL is vulnerable to privacy threats as discussed before.

Complementing governance is the pivotal area of Data Security. AI is only as good as the data it is fed. But this data, often sensitive and personal, needs to be shielded from unauthorized access and potential breaches. AI systems should not only prioritize users' privacy rights but should stand as Data Security fortresses. Ethical data acquisition is the starting point, ensuring that every piece of data is obtained with informed consent. Developers and operators of AI systems bear the responsibility of ethically managing this data throughout its lifecycle (Haakman et al., 2021). Their accountability extends beyond just Data Security, encompassing the outcomes that AI models produce. Regardless of intentions, developers and operators must take responsibility for the repercussions—be they beneficial or detrimental—that arise from AI's actions.

The narrative of governance and security is also intrinsically tied to Regulatory Compliance. As AI continues to influence every facet of our lives, the need to translate external AI regulations into actionable policies intensifies. However, Regulatory Compliance is a dynamic process. Ensuring that AI systems adhere to regulations requires meticulous design, rigorous testing, and ongoing evaluation. Beyond technical compliance, there exists an urgent need for a comprehensive legal and regulatory framework—one that not only ensures AI systems function within the legal boundaries but also upholds the rights of individuals. This framework embodies the principles of accountability and responsibility, ensuring AI's evolution harmonizes with societal norms and values.

In our exploration of AI, reproducibility, reliability, and communication principles emerge as foundational pillars. These elements converge to shape the broader narrative of Trustworthy AI systems. Reproducibility refers to the ability to consistently replicate AI research findings using the same data, methods, and experimental conditions, embodying the scientific rigor essential for advancing the field (Gundersen & Kjensmo, 2018). As AI's influence expands across diverse sectors, from healthcare to finance, the demand for repro-

ducible methods and generalizable results becomes paramount. The challenge is particularly acute in AI, where complex algorithms can obscure the path from primary evidence to derived knowledge, fostering a growing reliance on secondary sources or prevailing consensus. Yet, the essence of scientific inquiry lies in tracing information back to its primary origins and critically assessing its reliability. The rise of the "AI black box" risks limiting researchers' ability to scrutinize and correct errors, underscoring the necessity of reproducibility. Reproducibility, in contrast to explainability, does not aim to make models interpretable to humans, but rather ensures that the process and outcomes of AI systems can be consistently validated. Beyond verifying research, ensuring consistency in data, methodologies, and experiments, reproducibility strengthens trustworthiness by enabling the detection, analysis, and mitigation of risks in AI systems. Moreover, it accelerates progress by allowing the community to swiftly integrate cutting-edge approaches into practice or build upon them in follow-up studies (Li et al., 2023).

Reliability in AI systems refers to the probability that an AI model or software performs its intended functions without failure under specified conditions over a given period. Unlike traditional software, AI reliability is influenced by unique factors such as data quality, model accuracy, and environmental robustness. Failures in AI systems can stem from software errors (e.g., incorrect predictions due to adversarial attacks or dataset shifts) or hardware issues (e.g., sensor malfunctions in autonomous vehicles). The operating environment, including both physical conditions (e.g., temperature, lighting) and data conditions (e.g., training-test distribution mismatch), plays a crucial role in determining reliability (Hong et al., 2023).

A major challenge in ensuring AI reliability lies in the interplay between data, models, and environment. Poor data quality (e.g., imbalance, noise, or bias) can degrade model accuracy, while adversarial inputs or distributional shifts can lead to unexpected failures. Furthermore, uncertainty quantification is critical because AI predictions are probabilistic; high uncertainty can indicate unreliable outputs, particularly in safety-critical applications like healthcare or autonomous driving. Research in AI reliability must address these factors holistically, developing methods to assess failure risks, improve robustness, and establish trust in AI decision-making under real-world variability (Hong et al., 2023).

The significance of communication in AI cannot be overstated. It is not just about conveying information but fostering genuine understanding between AI systems and their users. As these systems grow in complexity, the duty falls upon developers, researchers, and communicators to demystify AI processes. While communication, transparency, and explainability in AI significantly overlap, all fostering user trust and understanding, they serve distinct purposes. Communication ensures clear and effective interactions between users and AI systems, encompassing output clarity (e.g., intuitive visualizations), engagement strategies (e.g., interactive interfaces), and AI literacy efforts (e.g., educating users about capabilities and limitations). Transparency and explainability, though often intertwined, play complementary roles. Transparency reveals the AI's inner workings, its design, decision-making processes, and potential biases, acting as a foundation for trust. Explainability goes further, justifying specific AI decisions in human-interpretable terms, whether through feature importance or technical methods like saliency maps. Together, they transform AI from a black box into a tool users can understand, interrogate, and ultimately rely on. This perspective on communication underscores the importance of transparency, user engagement, and fostering AI literacy. It is pivotal to understand not only how AI operates but also its origin, applications, and the logic behind its decisions. Fig. 1 demonstrates a brief overview of the AI principles and concepts discussed in sections 2 and 3.

## 4 Biases

AI bias refers to systematic prejudiced results produced by algorithms, which can lead to incorrect predictions. If harnessed appropriately, AI can satisfy expectations of accurate and fair decision-making. However, AI-empowered systems are vulnerable to biases that make their decisions unfair. Thus, focusing on ethical scrutiny of these systems is on the increase (Aquino et al., 2023).

Discrimination refers to a biased treatment of a certain group of people according to their age, gender, skin color, race, culture, economic conditions, etc (Calmon et al., 2017). While discrimination and bias are considered sources of unfairness, the former results from human prejudice and stereotyping and the latter is due to data collection, sampling, and measurement. Most AI systems are data-driven, thus, data plays a significant role in the functionality of these systems. Machine learning algorithms used for decision-making inherit any bias or discrimination in the training data. As a result, existing bias in data can produce biased

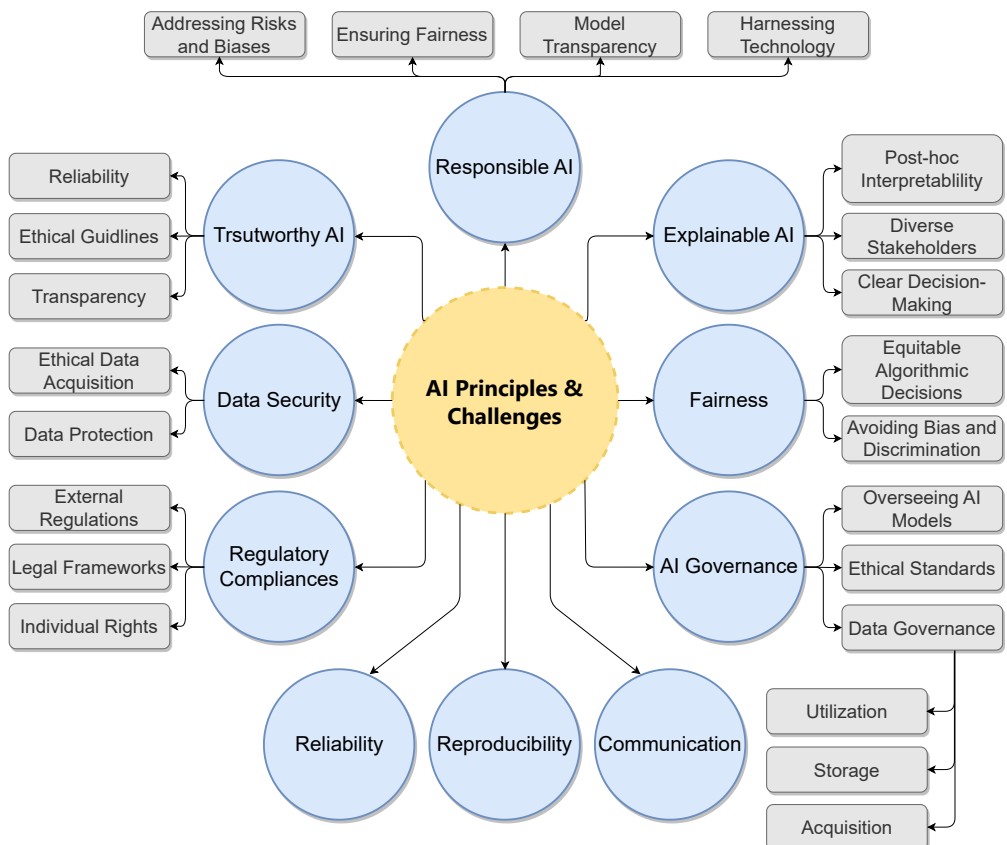

Figure 1: An overview of AI principles discussed in sections 2 and 3. The diagram depicts the intricate relationship and intersections between fundamental concepts crucial for the development and deployment of AI systems. Emphasized are nine central tenets: Responsibility, Explainability, Trustworthiness, Fairness, Data Security, AI Governance, Reliability, Reproducibility, and Communication. Surrounding these core ideas are various sub-components, serving as the building blocks and considerations that further refine and give depth to each principle, emphasizing the comprehensive nature of ethically implementing AI.

outcomes that are fed into real-world systems and can affect the end user's decisions (Anzum et al., 2022). In some cases, the algorithmic choice of design can also lead to biased outcomes, even if the data is not biased itself (Mehrabi et al., 2021).

It is essential to recognize and address the systemic social, political, and economic disparities that influence data and, consequently, the fairness of AI systems. For instance, unequal access to healthcare services, shaped by economic inequality or political exclusion, can lead to underdiagnosis or undertreatment of vulnerable populations. People of lower socioeconomic backgrounds or racial minorities often face barriers to adequate medical care, resulting in their underrepresentation in clinical datasets. This underrepresentation exacerbates bias in AI models and reinforces existing inequalities in healthcare delivery (Lekadir et al., 2021). Social determinants of health, such as income, education, and employment, are crucial factors in this context, especially for historically marginalized communities, including Indigenous Peoples, LGBTQ individuals, and Black Canadians. To counteract the impact of historical injustice and marginalization, positive discrimination may be warranted. These approaches deliberately prioritize data collection, inclusion, and tailored interventions for disadvantaged groups to ensure equitable representation and outcomes (Owens & Walker, 2020). Such strategies are not acts of bias but proactive steps toward fairness, aiming to restore balance where systemic inequities have long prevailed.

The heart-failure risk score that categorizes black patients as being in need of less care and cancer detection algorithms that perform poorly for people of color are two examples of racial discrimination and inequitable health practices for people of color (Owens & Walker, 2020). To ensure unbiased medical practices toward

patients of color, justice should be a fundamental principle in clinical and research ethics. This can be achieved by systematic education of health providers and applying anti-racist standards for the development and analysis of research (Owens & Walker, 2020).

The discussion on the socio-material character of data and how it reflects discrimination and historical inequalities is a critical conversation in social studies of science and technology. As we have seen in examples of AI generators trained with data over-represented for a determined population, it amplifies existing inequalities. The example of the images of doctors and nurses generated by MidJourney [1] is one of the most well-known cases. A critical question is how the propagation of images that reflect longstanding inequalities and discrimination reinforces users' bias concerning a determined issue.

There are three categories of biases, namely, data to algorithm, algorithm to user, and user to data that could cause a user feedback loop (Mehrabi et al., 2021). Firstly, 'data to algorithm' bias occurs when the data used to train AI systems is not representative or contains inherent prejudices. This can lead to AI algorithms that perpetuate and even amplify these biases in their outcomes. Secondly, there is a potential for 'algorithm to user' bias, where the decisions or perceptions of users are influenced by the biased outputs of AI algorithms. This interaction can subtly shape user behavior and judgments. Lastly, 'user to data' bias reflects the impact of human biases on user generated data, which, when used to train AI systems, can introduce or reinforce biases.

**Data to Algorithm** In ML algorithms, biases in data can be responsible for biased algorithmic outcomes. Methods of choosing, utilizing, and measuring certain features of a target group cause measurement bias. Cognitive bias, a subgroup of measurement bias, refers to the human brain's tendency to simplify information processing according to personal experience and preferences (Mehrabi et al., 2021). For example, if the database is generated by developers, engineers, or medical professionals in isolation, or by a group of people with a certain ethnicity or social background, chances are the database is unintentionally affected by their own biases (Ricci Lara et al., 2022). Omitting some variables from the model can also lead to a bias called omitted variable bias. Biases due to missing data information are usually more common within a specific population (prevalence of a specific population). If the data used to train the AI model does not adequately represent the diversity of human behavior, the system may be biased toward certain behaviors or groups. For instance, if the training data mainly consists of people from a certain geographical region, the model might not perform well or may misinterpret behaviors from people in other regions (Anzum et al., 2022).

Lack of diversity and proper representation of the target population in the training database is a type of data-driven bias (Anzum et al., 2022). Differences between demographics of the target population and the database, such as age, gender, ethnicity, etc., can lead to aggregation bias, ensuring that the model is not well-suited for all groups of the population (Anzum et al., 2022). For example, morbidity differences vary among diabetes patients with different ethnicities and genders. However, few available datasets contain clinical information and records such as age, race/ethnicity, gender, etc. Selection bias occurs if the sample selected for a study is not representative of the entire population (Yu & Eng, 2020). For example, if a study on social behavior only includes participants from a certain socio-economic background, it might not accurately reflect the behaviors of those from different backgrounds. Selection bias can take many different forms: Coverage bias, wherein data is not selected in a representative fashion; Non-response bias (or participation bias), wherein data is unrepresentative due to participation gaps in the data-collection process (Berg, 2005); Sampling bias, wherein proper randomization is not used during data collection (Jeong et al., 2018).

Data heterogeneity and target class imbalance can influence fairness and model performance in AI systems. Class imbalance refers to the unequal distribution of classes in the training dataset. Data heterogeneity can occur due to different equipment (vendor, model, etc.) and data acquisition protocols and the underlying distribution of subjects of a certain ethnicity, gender, or age (Dinsdale et al., 2021). Collecting data is a sensitive issue as it is susceptible to several biases (Acosta et al., 2022). Machine learning models are vulnerable to intrinsic characteristics of the data used and/or the algorithm employed. Any biases in data leads to biased outcome and spurious correlations that a specific model could exploit (Ricci Lara et al., 2022). Combining data from multiple sources or using multi-modal data, such as imaging data and electronic health records (EHR), can exacerbate the problem of bias (Acosta et al., 2022).

One significant solution to mitigate bias in the design, validation, and deployment of AI systems is to ensure diversity during data collection. This can be satisfied by enhancing transparency of datasets and

---

[1]https://www.midjourney.com

providing information about patient demographics and baseline characteristics (Vokinger et al., 2021). Data in healthcare typically includes patient demographics such as age, sex or gender, skin tone or race/ethnicity, and comorbidities. However some information might be absent due to privacy concern (Ricci Lara et al., 2022). Having a global perspective in the design of AI tools and applying meticulous approach to analyze the performance of these models based on population subgroups, including age, sex, ethnicity, geography, and sociodemographic status is of paramount importance (Goisauf & Cano Abadía, 2022).

Data labeling is a crucial aspect of the socio-technical system in which artifacts are designed and utilized. Most current AI models require human intervention for manually labeling data, a process that is often costly, complex, and time-consuming, especially when dealing with large volumes of data. One potential solution is to use semi-supervised or unsupervised learning algorithms. These methods allow the model to assign labels to unlabeled data, with human input sought only when the model's confidence is low. This approach can streamline the labeling process (Yakimovich et al., 2021). However, challenges remain, particularly regarding the potential for bias introduced by human involvement. People responsible for labeling are subject to organizational constraints and biased behavior, which may influence their decisions (Baker & Hawn, 2021). To mitigate this, clear, standardized methods for data collection and storage are necessary. Additionally, ensuring the fairness of AI systems involves considering standardized metadata and key variables like age, sex/gender, ethnicity, and geography during the data collection and preparation stages. These measures are essential for the integrity and effectiveness of AI models (Trocin et al., 2021).

**Algorithm to User** Algorithms affect user's behavior, thus, any bias in algorithms can lead to bias in the behavior of users. Algorithms have their own implicit biases even disregarding biases in data. Algorithmic bias refers to biases in the algorithms used to interpret data (Blanzeisky & Cunningham, 2021). If the algorithm is not properly designed or trained, it may incorrectly categorize or interpret certain behaviors, leading to unfair outcomes. For example, using DL models can augment original data bias due to model design (architecture, loss function, optimizer, etc.). Changes in population, cultural values, and societal knowledge can lead to the emergent bias. By using inappropriate benchmarks for the evaluation of an AI system, bias can arise during the evaluation process. For example, using benchmarks biased toward skin color or gender in facial recognition applications can create evaluation bias. Popularity bias occurs because popular objects appear more in public, thus they tend to be exposed more. Although popularity bias stems from skewed interaction data, its real impact occurs when algorithms deliver content to users. Algorithms trained on such data disproportionately favor already-popular items, leading to feedback loops where these items gain even more exposure and this biased decision-making directly shapes user experience and behavior (Mehrabi et al., 2021).

**User to Data** Many of databases are generated by human, therefore, any inherent biases in the behavior of users might introduce bias into data sources. Further, due to the impact of algorithms on the behavior of users, algorithmic biases can result in data bias. In cases where statistics, demographics, representatives, and user characteristics in end users are different from those of the original target population, population bias occurs. Demographics include age, sex, gender, race, diverse genotypes, ethnicity/genetics, and appearance. Self-selection bias, which is a subtype of the selection bias, occurs when subjects of research select themselves. Behavioral bias refers to the different behavior of users across different platforms or datasets. The differences in behaviors over time is a temporal bias (Mehrabi et al., 2021).

Here, other sorts of bias, including visible minority bias, research bias, and group attribution bias are introduced. Exclusion of patients with rare diseases and disabilities, such as rare genetic variance, from training data can affect the performance of AI systems. This is due to sampling bias and lack of generalizability. Neglecting patients with rare diseases or disabilities from research on the application of AI in disease diagnosis can have significant negative implications for patients and AI developers and affect the trustworthiness of AI in clinical settings (Hasani et al., 2022).

Racial minority groups suffer from increased risk of diagnostic error, resulting from unequal access to healthcare and disparities in the quality or delivery of diagnostic systems. Moreover, in the clinical context, algorithmic bias in AI systems has worsened the existing inequity in healthcare and as a result, under-represented and marginalized groups are disadvantaged (Aquino et al., 2023). The lack of research and available data on marginalized (under-served) communities such as intersex, transgenders, and LGBTQ2S+ have raised bias for this minority population. Few studies and lack of enough data from them, due to narrow and binary background assumptions regrading sex and gender, has worsened healthcare inequity. As another example,

effects of sex and gender dimensions on health and diseases have been overlooked by AI developers. Indeed, gender/sex imbalanced datasets contribute to bias in model performance and disease diagnosis. Thus, it is recommended to not only include under-served communities in research to mitigate bias but also to introduce desirable bias to counteract the effects of undesirable bias and discrimination (Goisauf & Cano Abadía, 2022). Another example in this context is the application of biometric systems trained on datasets mainly from one culture. The aim of biometric systems is to establish or verify demographic attributes, such as age, race, and gender.

One of the most significant topics in this context is research biases. In low-income countries, lack of research funding prevents conducting research on many health problems which raises bias against some ethnicities and increases health inequity. Implicit bias occurs when assumptions are made based on one's own mental models and personal experiences that do not necessarily apply more generally. A common form of implicit bias is confirmation bias, where model builders and researchers consciously or unconsciously seek data or interpret results in ways that affirm preexisting beliefs. In some cases, a model builder may actually keep training a model until it produces a result that aligns with their original hypothesis; this is called the experimenter's bias. Confirmation bias leads to biased conclusions and increases bias in research. By hiring a multidisciplinary team of experts and stakeholders, including AI developers, medical professionals, patients, and social scientists, and applying their perspectives to the design, implementation, and testing of AI algorithms, one can improve fairness in research.

One very important component relevant to the research bias and confirmation bias, particularly in the health space, is how to decide what is "authoritative" - the basis for interpreting the often conflicting and always noisy information in the literature. For example, there is a tendency to rely on the Evidence-Based Medicine (EBM) structure and evidence ranking criteria when assessing the medical literature. The EBM structure considers applying contemporary scientific evidences in the decision-making and treatment planning of each individual patient (Bluhm, 2005). EBM is often presented in very definite terms, but ultimately it is a useful heuristic to assist in decision-making by non-specialists (Solomon, 2011). EBM has some limitations due to bias in the reporting of clinical trials, as a result of subject selection (Rawlins, 2008), model performance, and analysis of results, as well as journals' reluctance to publish negative results (Sheridan & Julian, 2016). It is quite likely that AI researchers collaborating with clinical experts, particularly if no specialists with basic research training are involved, might wind up adopting EBM approaches without consideration of the assumptions embedded in them. Any system trained on the basis of this approach risks setting in stone of biases, in ways that may be completely opaque to the end user (Greenhalgh et al., 2014).

Although research retraction should be managed and resolved immediately, the process takes years. This will propose challenges for EBM and AI, especially in medicine. Indeed, even when a paper is retracted, flawed data used for training and validation will exist on the Internet for years, affecting workflows, analysis, and model performance of AI systems. Notwithstanding the significance of online data for advancement in AI-based systems in various fields, certainty of the quality, reliability, and annotation of data should be taken seriously to settle the issue of retraction in research (Holzinger et al., 2022).

Group attribution bias is a tendency to generalize what is true of individuals to an entire group to which they belong. Two key manifestations of this bias are In-group bias and Out-group homogeneity bias. In-group bias refers to a preference for members of a group to which they also belong, or for characteristics that they also share. Out-group homogeneity bias refers to a tendency to stereotype individual members of a group to which they do not belong or to see their characteristics as more uniform (Böhm et al., 2020).

## 4.1 Strategies to Detect Biases

Although most effort (particularly at the bench) is focused on prospectively avoiding biases (some studies are done on tissue from both male and female donors, diverse genotypes, etc.), bias in data and algorithms is inevitable. The impact of biases is not completely understood and may not be apparent until very late in the process (sometimes after a drug has entered trials). AI may be useful here in flagging specific concerns for researchers at earlier stages - for example, if a researcher is working with a specific pathway, and a minority group within the population is known to have genetic variation in genetic loci in the vicinity of one of the genes involved, the researcher may not be aware of that as there is simply too much information for any one person to keep track of.

Algorithmic fairness focuses on designing fair algorithms to reduce bias in machine learning. Two main approaches include awareness-based and rationality-based fairness. Awareness-based fairness consists of fairness through awareness and fairness through unawareness. Fairness through awareness means that individuals who are similar based on all relevant attributes, including sensitive ones, should receive similar treatment. This method relies on a distance metric that defines how similar individuals are, making the algorithm "aware" of these relationships. On the other hand, fairness through unawareness removes sensitive features from the decision process to avoid bias. However, this can still lead to indirect discrimination if other features are correlated with the sensitive attribute. Rationality-based fairness includes statistical definitions, like demographic parity and equal opportunity, as well as causal approaches, like counterfactual fairness (Wang et al., 2022). In summary, these fairness approaches aim to guide algorithmic decisions toward equitable outcomes by either recognizing or excluding sensitive attributes.

Detecting data bias and algorithmic bias is a sensitive issue as it can affect the trustworthiness of AI systems. There exist different methods to detect bias in data and algorithms. Disparate Impact Analysis is a quantitative method used to detect bias in AI systems (Zafar et al., 2017). It measures how the system's outcomes differ across different demographic groups. For instance, if an AI system makes accurate predictions for one group but not for others, there may be a bias in the system. Bias Audit is a comprehensive method for the evaluation of the AI system using various metrics. It can involve reviewing the data used to train the model, examining the algorithm's decision-making processes, and testing the system's outcomes with different input data. Counterfactual Analysis involves changing the features of the data that represent protected characteristics (like race, gender, etc.) and assessing changes in the AI system's output. Significant changes could be an indication of bias. (Mothilal et al., 2020).

Agarwal et al. (Agarwal et al., 2018) proposed a test generation technique that detects all combinations of protected and non-protected attributes in the system that can cause bias. Srivastava and Rossi (Srivastava & Rossi, 2018) proposed a two-step bias detection approach, wherein a 3-level scale bias rating is developed to identify whether an AI system is unbiased, data-sensitive biased, or biased. To detect statistical and causal discrimination at the individual level, Black et al. (Black et al., 2020) proposed a fairness testing approach called the Flip Test.

Researchers have developed fairness toolkits in order to ease bias detection and mitigation. Salerio et al. (Saleiro et al., 2018) proposed a bias detection toolkit to identify and correct data bias before training the model. There exist some toolkits to address bias detection and bias mitigation as well. Bantilan et al. (Bantilan, 2018) proposed a toolkit named Themis-ML to detect and mitigate bias. This technique provides fairness metrics, such as mean difference, for bias detection, and relabeling and additive counterfactual fair estimator for bias mitigation. Bellamy et al. (Bellamy Rachel et al., 2019) introduced an extensible toolkit called AI Fairness 360 to detect, understand, and mitigate algorithmic bias. The AI Fairness 360 is a comprehensive toolkit that brings together a thorough set of bias metrics for bias detection, bias algorithms, and a unique extensible metric explanation facility to provide end users with the meaning of bias detection results. Fairlearn is a Python library designed to help AI developers evaluate and improve the fairness of their machine learning models. It provides tools to measure fairness metrics and includes algorithms to address and reduce biases in AI systems (Weerts et al., 2023). Google's What-If Tool (WIT) is an interactive, open-source interface for probing machine learning models to assess fairness, performance, and behavior across different data subgroups. It allows users to visualize model predictions, test counterfactuals, and analyze bias without writing code, supporting models from TensorFlow, scikit-learn, and XGBoost. WIT is particularly useful for debugging classification and regression models in NLP, computer vision, and tabular data (Wexler et al., 2019).

## 4.2 Bias Mitigation Techniques

Researchers focusing on Trustworthy AI have offered several methods to address bias and make AI systems fair. Bias mitigation techniques are categorized in three main classes, namely, Pre-processing, In-processing, and Post-processing (Mehrabi et al., 2021). The Pre-processing method deals with modifying the training data, in case it is allowed by the algorithm, to remove any bias and discrimination, assuring the data is unbiased, mitigating over- or under-representation of any specific population. In In-processing techniques, by changing and modifying ML algorithms during the training process, bias can be removed. Post-processing algorithms can be applied to the model predictions to remove bias (Mehrabi et al., 2021).

Several Pre-processing algorithms have been proposed by researchers. Feldman et al. (Feldman et al., 2015) proposed a technique that hides protected attributes from the training dataset while preserving the data properties. Calmon et al. (Calmon et al., 2017) proposed a probabilistic framework based on an optimization formula to transform data in order to reduce algorithmic discrimination. Samadi et al. (Samadi et al., 2018) proposed a linear dimensionality reduction technique to reduce bias in the training data. Kamiran and Calders (Kamiran & Calders, 2012) proposed three data Pre-processing techniques to remove bias from the training data, namely, Massaging technique, Reweighing technique, and Sampling technique.

Several research have been conducted to mitigate bias in algorithms during the training process. Huang and Vishnoi (Huang & Vishnoi, 2019) proposed a stable and fair extended framework based on fair classification algorithms to mitigate bias in the classification process. They applied a stability-focused regularization term to guarantee the stability of their proposed approach with respect to variations in the training dataset. Bechavod and Ligett (Bechavod & Ligett, 2017) proposed a penalty term as a regularization, which is data-dependent and set at the group level to mitigate bias during the learning phase.

Various Post-processing approaches are proposed to mitigate bias using the output of the predictors. Menon and Williamson (Menon & Williamson, 2018) investigated the classification method which is appropriate to trade off accuracy for fairness. The optimal classifier for cost-sensitive approximation fairness measure is an instance-dependent thresholding of the class probability function. Also, they suggested that by measuring the alignment of the target and sensitive variable the degradation in performance can be quantified. Dwork et al. (Dwork et al., 2018) proposed an efficient decoupling technique to be added on top of the ML algorithms. To mitigate bias, this method uses different classifiers for different groups.

In spite of recent advancements in bias detection and mitigation algorithms, Ad hoc attempts to identify biased datasets should not be neglected. When exploring data, missing or unexpected feature values that stand out as especially uncharacteristic or unusual should be considered seriously. These unexpected feature values could indicate problems that occurred during data collection or other inaccuracies that could introduce bias. Further, any sort of skew in data, where certain groups or characteristics may be under- or over-represented relative to their real-world prevalence, can introduce bias into the model.

### 4.3 Evaluating Biases

Evaluation bias occurs if the wrong evaluation metric is applied to report the performance of a model. Hence, selecting appropriate fairness evaluation metrics to detect different types of bias in the system is highly significant. The development and deployment of Trustworthy AI requires quality assessment and risk management to enhance its reliability. Bias in AI systems can lead to unfair and unreliable decisions. Hence, risk management in AI systems should be taken into account to ensure fair decision-making.

There are some suggestions to assist risk management. First, model facts and workflows should be automated for compliance with business standards. Identifying, managing, monitoring, and reporting on risk and compliance at scale is highly recommended. Human agency and oversight are required for Trustworthy AI. Humans should be responsible for developing algorithms, setting limits for performance, identifying and correcting errors in the system, and improving the performance of the system by giving continuous feedback. Dynamic dashboards should be utilized to provide customizable results for stakeholders. This is important for different stakeholders using or impacted by AI systems to clearly understand the performance and limitations of these systems. Enhancing collaboration across multiple regions and geographies is another factor to reduce bias risk in AI systems. This can increase the diversity of data which leads to unbiased and fair training data.

The application of AI systems in various domains is on the increase. However, bias risks in these systems have raised unreliability and prevented users from completely accepting and trusting these systems. Thus, fairness evaluation metrics are highly recommended to increase acceptance and trust for AI-based decision-making systems. These metrics should be computed in the entire set as well as different sets, to check model performance for each group. Here, some of the main fairness evaluation metrics are tabulated in Table 1 to detect bias and improve Fairness in AI systems.

For instance, fairness evaluation metrics used for biometric systems are discussed here. In the biometric face recognition domain, accuracy is measured based on the Signal Detection Theory (SDT). In this technique, pairs of images of the same person or two different people are compared (Mukhopadhyay, 2016). The output is a similarity score. Accuracy is measured based on the degree of overlap between the similarity score

Table 1: Evaluation Metrics for Bias Detection

| Fairness Evaluation Metrics | Definition | Formula |
|---|---|---|
| Equality of Accuracy across groups | The aim is to ensure that both privileged and unprivileged groups have equal prediction accuracy | $P(\widehat{Y} = Y \mid z = 0) = P(\widehat{Y} = Y \mid z = 1)$, z=0 and z=1 are unprivileged and privileged groups, respectively. $\hat{Y}$ and Y represent model predictions and the ground truth, respectively. |
| Disparate Impact (DI) | The aim is to compare the proportion of individuals receiving a positive outcome between the unprivileged group and the privileged group. | $DI = \min\left(\dfrac{P(\widehat{Y}=1\mid z=0)}{P(\widehat{Y}=1\mid z=1)}, \dfrac{P(\widehat{Y}=1\mid z=1)}{P(\widehat{Y}=1\mid z=0)}\right)$ |
| Group Unawareness | A model's decision would be fair if it is unaware of sensitive attributes. | It is equivalent to feature attribution in ML. The attribution of protected attributes should be 0. |
| Demographic Parity | The aim is to ensure that different demographic groups experience equal proportions of positive outcomes. In other words, this measurement ensures that the model prediction is statistically independent of the protected attribute. | $P(\widehat{Y} = 1 \mid z = 0) = P(\widehat{Y} = 1 \mid z = 1)$ |
| Equal Odds | The aim is to ensure that privileged and unprivileged groups have equal True Positive Rate (TPR) and equal False Positive Rate (FPR). | $TPR = \dfrac{TP}{TP+FN}$, $FPR = \dfrac{FP}{FP+TN}$, where TP, FN, FP, and TN are True Positive, False Negative, False Positive, and True Negative, respectively. |
| False Positive Rate (FPR-Parity) | The aim is to ensure that all groups have the same FPR. | |
| Equal opportunity Or True Positive Rate (TPR) | The aim is to ensure that decisions are not based on protected attributes but rather on qualifications and metrics and that all groups have equal access to opportunities, regardless of demographics. | $EO = 1 - |TPR_{z=0} - TPR_{z=1}|$ |
| Positive Predicted Value (PPV-Parity) | The aim is to ensure that the chance of success is equal based on a positive prediction. | $PPV = \dfrac{TP}{TP+FP}$ |
| Negative Predicted Value (NPV-Parity) | The aim is to ensure that the ratio of correctly rejecting samples out of all the samples the model has rejected is the same for each group. | $NPV = \dfrac{TN}{TN+FN}$ |
| Predictive Value Parity | It is satisfied when both PPV and NPV are satisfied. | |

distributions and it is summarized by the receiving operating characteristic (ROC) curves and synopsized as the area under this curve(AUC) (Zhang et al., 2015). In applications, a threshold similarity score is generated to measure additional accuracy measurements and evaluation of the identification of different demographics or genders, including False Accept Rate (FAR), False Rejection Rate (FRR), Equal Error Rate (EER),

precision, recall, and other metrics. The EER, FAR, and FRR can be applied for face recognition, biometric identity recognition, and emotion detection data biases (Sundararajan et al., 2019).

As another example, we investigate fairness evaluation metrics for loan approval. In the process of loan approval, financial and credit details of borrowers are verified by lenders to decide whether to approve or deny their request. In this context, bias can be evaluated through historical data and fairness across locations/populations. Various methods can be applied here to detect bias, including statistical parity, conditional demographic parity, individual fairness, and counterfactual fairness. Conditional demographic parity can be applied to calculate the potential existence of indirect discrimination in the loan approval process (Genovesi et al., 2023).

## 4.4 Tracing and Mitigating Bias in Intelligent Systems

Here, we summarize the most important sources of bias and explain a process through which we can develop Fairness in AI systems (Fig. 2). In this figure (Fig. 2), we summarized the feedback loop of bias, in which bias generated by each source is circled back and enters into another source as input bias. According to Fig. 2, bias can intrude on AI systems during data collection and/or model development processes. Thus, applying fairness evaluation metrics is needed to detect and mitigate bias in order to improve Fairness in AI systems.

A number of fairness evaluation metrics are presented in Fig. 2. Assessing bias depends on whether the research is empirical-based or more theoretical. Philosophers in conjunction with social scientists came up with both qualitative and quantitative measures of assessing biases in AI design. We can consider various evaluation metrics to detect bias. Bias can be detected mostly through quantitative standard metrics (e.g., accuracy across protected groups) to detect a problem and Explainable AI techniques (e.g., saliency maps or counterfactual images) to find out what might be causing them. Causal analysis is also getting more and more popular to identify (and potentially mitigate) problems. There are several fairness evaluation metrics in ML, such as demographic parity, equal opportunity, and disparate impact, which can be used to quantify the extent to which an AI system is biased. Some of these metrics can be assessed through elements of the confusion matrix, such as True Positive Rate parity. Confusion matrix is a table layout that allows visualization of the performance of an algorithm. This matrix can show the number of False Positives, False Negatives, True Positives, and True Negatives, which can help detect any bias in predictions. Bringing in third-party experts to review the AI system can be helpful in identifying bias. These experts can scrutinize the design of the system, the data used to train it, and its decision-making processes to find any potential sources of bias.

In this figure (Fig. 2), the process of applying bias mitigation techniques is illustrated. Some techniques can be applied in the Pre-processing stage to mitigate bias in the training data, such as reweighing and disparate impact remover (DIR). Reweighing assigns higher weights to training samples from unprivileged groups to balance the representation (Kamiran & Calders, 2012). DIR reduces the correlation between features and sensitive attributes, ensuring equal feature distributions across protected groups (Feldman et al., 2015). Other bias mitigation methods operate during the In-processing stage, targeting bias during model training. One example is adversarial debiasing, which uses an adversarial framework with two components: a predictor that learns to predict the target variable accurately, and a discriminator that attempts to infer the sensitive attribute from the predictor's outputs. The predictor is trained to minimize both prediction error and the ability of the discriminator to detect bias (Zhang et al., 2018). Finally, bias mitigation can be performed in the Post-processing stage, after detecting bias in model outputs. calibrated equalized odds (CEO) is one such technique; it modifies the predicted probabilities of a calibrated classifier and adjusts output labels to achieve equalized odds (Pleiss et al., 2017).

Despite the effectiveness of bias mitigation techniques in fairness improvement, a significant challenge lies in the accuracy-fairness trade-off. One of the main causes of this trade-off between accuracy and fairness is due to the high correlation between sensitive attributes and target variables. Indeed, when target variable and sensitive features are highly correlated, more significant adjustments to the model are required to achieve fairness. Thus, improving fairness can lead to greater reduction in accuracy because fairness is achieved by removing correlations between sensitive features and target variable (Menon & Williamson, 2018). Further, by enforcing fairness constraints, such as equalizing False Negative Rates, the model is compelled to treat different groups similarly, even when there is a significant difference between groups. This can lead the model

to overlook the most dominant patterns in the data that are highly important for predictions, which can lead to reduced accuracy (Buijsman, 2024).

Whereas bias detection and mitigation during data collection and model development is of utmost importance, the significance of post-authorization monitoring should be taken seriously. Monitoring the performance of an AI system after deployment is an essential factor to avoid unwanted bias. At this stage, bias can arise due to data used for testing and system applied for practice. For example, in clinical practices, when the patient cohort differs from that in the training data, an AI system deployed suffers from heterogeneity in data and may encounter domain shift. Domain shift is due to the distribution difference between training and test datasets. Qualitative user studies involve collecting qualitative data from users of the AI system through direct observation and study of participants to identify the experience of users and gain insight into problems that exist within a service or product. The output of qualitative studies is the motivation, thoughts, and attitudes of people observed. For example, participants can provide insights into how an AI system is perceived by different demographic groups and whether they feel it is fair and unbiased. Cross-platform comparison (using different social media platforms) can be applied to obtain demographics and population samples for qualitative user research. Also, we can stick to common ML techniques and evaluate per-class or per-group fairness evaluation metrics.

## 5 Trustworthy and Responsible AI in Human-centric Applications

In recent years, researchers exploring diverse applications of AI in human-centric decision-making intelligent systems have directed their efforts towards a deeper understanding and identification of biases, along with the development of techniques to mitigate these biases. Additionally, there is a focus on enhancing the efficiency of methods for evaluating AI models, with specific attention to trustworthiness, explainability, fairness, and interpretability. We categorized the literature review into different subsections according to different human-centric applications, including healthcare, medical imaging, human-robot interaction, autonomous vehicles, philosophy and social sciences, biometric and cybersecurity, education, and social media. Summary of the main research advancements in those application domains is shown in Table 2.

*Healthcare:* healthcare is one of the domains where quality and safety issues in medical AI research are of paramount importance. Challen et al. (Challen et al., 2019) proposed a general framework for considering clinical AI quality and safety issues, containing several short-term, medium-term, and long-term issues. For example, distributional shift, black box nature of AI systems, and deployment of systems with no confidence in their prediction accuracy present short-term issues for medical AI. The proposed framework is supported by a set of quality control questions for short-term and medium-term issues to help clinical safety professionals and ML developers to recognize areas of concern. Trocin et al. (Trocin et al., 2021) proposed a systematic literature review for data collection and sampling, aiming for intellectual structure analysis of Responsible AI in digital health. In this research, various definitions related to Responsible AI are discussed and ethical aspects of implementing AI in healthcare were investigated. Moreover, a research agenda is proposed for future research in developing Responsible AI for digital health, with six ethical concerns, namely inconclusive evidence, inscrutable evidence, misguided evidence, unfair outcomes, transformative effects, and traceability. Vokinger et al. (Vokinger et al., 2021) conducted research on mitigating bias in ML in clinical settings. A number of steps are introduced to mitigate bias during data collection and preparation. To mitigate sampling bias, transparency of training data, using diverse and large training data, and monitoring and detecting errors of ML systems are of utmost importance. Badal et al. (Badal et al., 2023) proposed a guiding principle for Responsible AI for healthcare. In this study, eight principles were offered for AI developers and breast cancer was taken as an example to investigate the importance of these ethics in healthcare. According to the principles, AI should be developed to improve health equity, healthcare values, and generalizability. It was also indicated that AI systems should be enhanced in terms of explainability and interpretability, which enables them to contribute to the improvement of healthcare systems.

Mittemaier et al. (Mittermaier et al., 2023) investigated bias in AI-empowered systems for medical applications and discussed a strategy to mitigate bias in surgical AI systems. Authors indicated that bias in AI models can be generated in different stages, including data collection and preparation, model development and evaluation, and deployment. Bias mitigation strategies can be applied in three stages, namely, Pre-processing stage, In-processing stage, and Post-processing stage. Implementing AI for medical applications can raise challenges regarding bias, data heterogeneity, and continuous update of AI models to detect

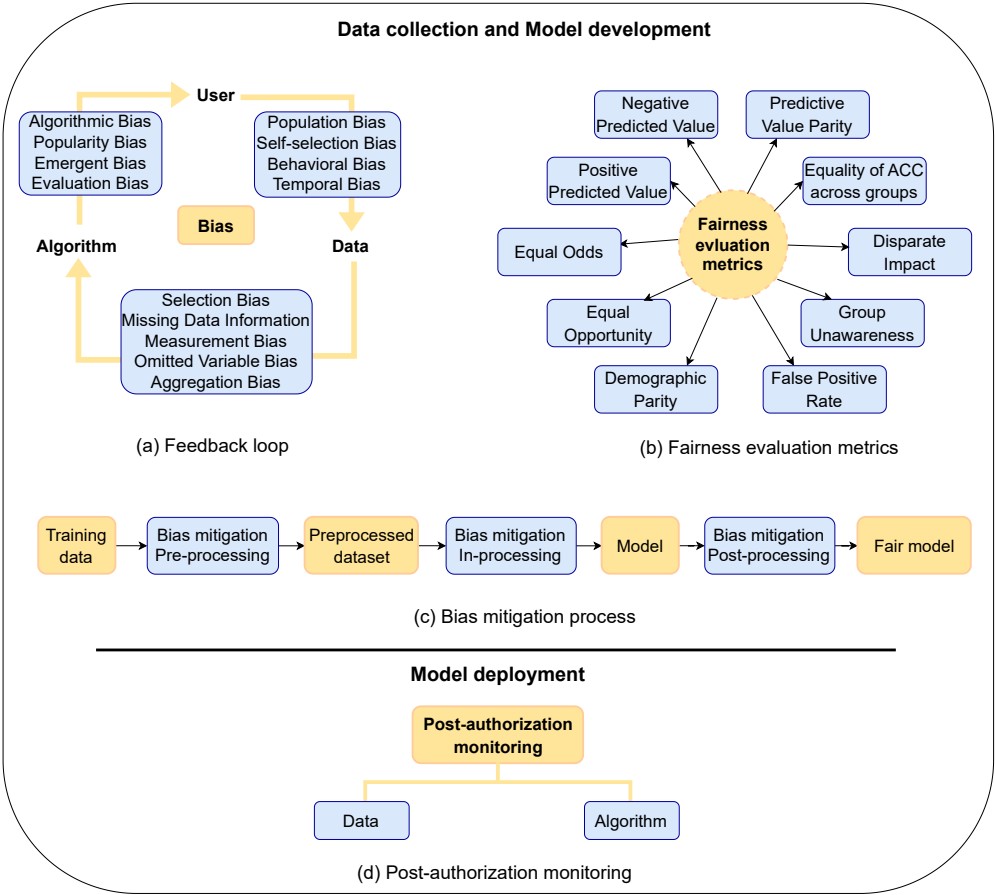

Figure 2: The summary of sources of bias, bias detection, and bias mitigation techniques. a) the feedback loop of bias, wherein various biases that can intrude on AI systems during data collection and model development are depicted. b) a number of fairness evaluation metrics that can be applied to identify bias in AI systems. If the bias detected can be addressed, the model should be adjusted. For each specific problem one or more metrics can be applied in the Pre-processing, In-processing, and Post-processing stages. c) a design process to mitigate bias in the Pre-processing, In-processing, and Post-processing stages to improve Fairness in AI systems. Examples include Pre-processing methods such as reweighting and disparate impact remover; In-processing methods like adversarial debiasing; and Post-processing methods like calibrated equalized odds post-processing.
d) post-authorization monitoring of data and AI system, through qualitative user studies or fairness evaluation metrics.

and mitigate bias arose from new data. Albahri et al. (Albahri et al., 2023) proposed a systematic review of Trustworthy AI in healthcare. The papers were categorized into seven categories, including explainable robotics, prediction, decision support, blockchain, transparency, and recommendations. It was indicated that Trustworthy AI can contribute to the enhanced clinical understanding of the diagnosis, improved patient care, protected privacy, assistance of health providers, and developed principles and global collaboration. Zhang et al. (Zhang et al., 2023) investigated four features related to Responsible AI that prevents harnessing the full potential of AI-empowered tools in clinical settings. In this paper, factors that affect the responsibility and trustworthiness of AI tools in clinical practices were discussed, including data and algorithmic bias, as well as the black box nature of AI-empowered models. Authors mentioned that, poor contextual fit, such as overlooking clinicians' workflow and collaboration in clinical settings, can pose challenges for deployment of AI systems in healthcare. Giovanola and Tiribelli (Giovanola & Tiribelli, 2023) explored the idea of fairness in healthcare machine learning algorithms (HMLA) and highlighted the ethical significance of fairness beyond

simple non-discrimination and lack of bias. The authors argued for a broader understanding of fairness that includes both distributive and socio-relational dimensions, emphasizing respect for individuals as unique persons.

Ueda et al. (Ueda et al., 2024) characterized fairness in healthcare as a multidimensional concept aimed at ensuring equitable distribution of resources, opportunities, and outcomes across diverse patient populations. They identified two primary sources of AI bias - data bias and algorithmic bias - while noting healthcare-specific biases may also emerge. Clinician interaction-related bias occurs when healthcare professionals engage with AI systems, potentially affecting algorithmic performance, fairness, and adaptation. Additionally, patient interaction-related biases may arise from unequal distribution of AI-driven healthcare benefits, potentially exacerbating existing health disparities.

*Medical Imaging:* Medical imaging lies at the intersection of healthcare diagnostics and computer vision domains. Ricci Lara et al. (Ricci Lara et al., 2022) investigated potential sources of biases in AI systems for medical imaging and explored strategies to mitigate them. According to this study, AI systems can be biased against certain sub-populations by attributes such as age, gender, socioeconomic status, and ethnicity, among others. Authors also indicated that data, model design, and people developing AI systems contribute to these biases. They also introduced some strategies to mitigate biases before, during, and after training models.

Hasani et al. (Hasani et al., 2022) investigated medical, ethical, and societal aspects of using AI for the Rare Disease (RD) treatment. Although AI plays a critical role in diagnosing and managing RDs, some ethical considerations should be noticed when using these systems, such as eliminating data of patients with RDs at the training stage, stigmatization, and revealing incidental findings. In this study, different ways to enable developments in RD diagnosis and treatment using AI tools were explored. According to this research, building trust in using AI for patients and healthcare systems, as well as developing collaboration between developers and medical professionals is essential to enhance knowledge in the application of AI in RDs diagnosis and treatment. Bernhardt et al. (Bernhardt et al., 2022) investigated dataset biases, namely population shift, prevalence shift, and annotation shift, and their effects on the AI model.

Stanley et al. (Stanley et al., 2023) proposed a systematic framework to investigate the impact of bias on medical imaging AI. A bias generation tool, called Simulated Bias in Artificial Medical Images (SimBA), was applied to evaluate the performance of the framework for bias mitigation. The proposed bias mitigation techniques included reweighing, bias unlearning, and group models. Jones et al. (Jones et al., 2023) highlighted the importance of understanding the mechanism of dataset bias in order to choose appropriate mitigation strategies and fairness metrics. They identified three types of bias mechanisms in clinical settings, namely, prevalence disparities, presentation disparities, and annotation disparities and provided a three-step reasoning to ensure fairness in medical imaging and AI-based decision-making systems.

Xu et al. (Xu et al., 2024) proposed a systematic review to examine fairness challenges in deep learning-based medical image analysis, identifying data bias and algorithmic bias as primary sources of inequity. The authors categorize mitigation approaches into pre-processing, in-processing, and post-processing techniques, while highlighting domain-specific hurdles such as labeler subjectivity in radiology and heterogeneous imaging protocols across institutions. Critically, the review reveals that most current methods prioritize group fairness metrics without addressing clinical utility trade-offs or causal determinants of health disparities, underscoring the need for clinician-AI collaboration frameworks to align technical fairness with equitable patient outcomes.

*Human-Robot interaction:* Human-robot interaction (HRI) is a prominent example of AI systems embedded in human-facing socially interactive contexts. As robots increasingly collaborate with people in personal, professional, and public environments, designing for trust becomes essential—not only in terms of technical reliability but also through behavior that aligns with human expectation, supports mutual understanding, and adapts to the nuances of interaction.

To investigate the importance of transparency in AI-based systems, a study (Mercado et al., 2016) examined how the level of transparency in an intelligent agent (IA) impacts the performance, trust, and workload of operators involved in managing multiple unmanned vehicles (UxVs). Participants acted as UxV operators, completing missions by giving orders through a computer interface. The IA provided recommendations

for each mission, with varying levels of transparency. Results showed that higher transparency improved operator performance, trust, and perceived usability. Workload did not increase with transparency, and response times remained consistent across transparency levels.

Cantucci and Falcone (Cantucci & Falcone, 2020) proposed a cognitive architecture that enables social robots to build user profiles and apply a theory of mind for adaptive task delegation. This framework promotes explainability by allowing the robot to articulate its reasoning during interaction and enhances transparency. Similarly, Hayes and Shah (Hayes & Shah, 2017) implied that when users are exposed to understandable robot control policies, they calibrate trust more effectively and collaborate with greater confidence. Extending these insights, Chi and Malle (Chi & Malle, 2023) demonstrated that trust develops more robustly when users are actively involved in teaching and guiding robot learning—suggesting that co-adaptive engagement fosters resilience even in the presence of system errors.

Emerging perspectives also highlight trust as a socially and organizationally situated phenomenon. Kopp (Kopp, 2024) observed that trust and distrust can coexist in collaborative settings, functioning as parallel strategies for managing uncertainty. In such environments, trustworthiness is shaped by performance and how the system fits into existing roles and expectations. Moreover, Green and Iqbal (Green & Iqbal, 2025) explored how the robot could sense user trust level through multimodal signals such as gaze, facial expressions, and physiological cues. By allowing robots to respond to subtle indicators of discomfort or disengagement, these mechanisms could ensure a responsive and socially tuned interaction. Overall, these advances suggest that fostering trust in HRI involves both intepretability and adaptivity —where systems learn with users, signal intentions clearly, and behave in ways that are contextually and relatively coherent.

*Autonomous Vehicles:* In smart city applications, such as autonomous vehicles (AVs), deploying ML technology can raise concerns about algorithm complexity, transparency and the potential for misleading classifications. Therefore, in the context of AVs, understanding and fostering trust are pivotal for widespread adoption. Koo et al. (Koo et al., 2015) emphasized the delicate balance between explainability and cognitive overload, cautioning that excessive explanation might reduce rather than improve user trust. On the ethical front, Lim and Taeihagh (Lim & Taeihagh, 2019) delved into algorithmic decision-making in AVs, identifying biases stemming from statistical discrepancies in input data, legal and moral standards, and a lack of accountability frameworks. Similarly, Soares et al. (Soares et al., 2019) proposed an interpretable neuro-fuzzy model with a density-based feature selector to generate transparent driving decisions—validated using real-world data from Ford Motor Company (Nageshrao et al., 2019).

Investigating trust in AVs, Zhang et al. (Zhang et al., 2020) applied the expectation-confirmation theory, finding that trust is increased when perceived performance exceeds expectations. Addressing these concerns, Yu et al. (Yu et al., 2020) introduced the "BDD100K" dataset, emphasizing geographic, environmental, and weather diversity for mitigating decision-making bias. Meanwhile, Katare et al. (Katare et al., 2022) focused on bias detection in AI models for autonomous driving. The authors stressed the necessity of diverse object distributions in training/testing sets using the biased-car dataset (Madan et al., 2021) and employed selectivity score and cosine similarity on the nuScenes dataset (Caesar et al., 2020) for robust detection of biases in pedestrian and car recognition. This nuanced exploration encompasses both the intricacies of trust and the ethical dimensions surrounding AVs. In another research, Asha and Sharlin (Asha & Sharlin, 2023) conducted participatory interview sessions with senior participants to design interactions with AVs and discovered that an overload of instructions and notifications during AV interactions can escalate anxiety and frustration.

Recent research continues to refine how trust can be fostered through adaptive interaction and user-aware explanation strategies. Ding et al. (Ding et al., 2024) demonstrated that critically adaptive displays, which emphasize road-relevant hazards based on urgency and context, can significantly enhance user trust and situational awareness—particularly when tuned to individual characteristics such as trust propensity and age. From a complementary perspective, Ling et al. (Ling et al., 2024) integrated uncentainty-aware explanations into AV perception systems, showing that when a vehicle trasperantly communicates its confidence in environmental interpretations, users form more calibrated and appropriate trust responses. These findings reinfornce that trustworthy AI in AVs is not solely a matter of accuracy or automation level—it is about enabling intellibility, responsiveness, and alignment with human expectations in dynamics,

safety-critical environments.

*Philosophy and Social Sciences:* Mittelstadt et al. (Mittelstadt et al., 2019) conducted research on the difference between Explainable AI and explanations in philosophy, cognitive science, and social science. They argued that developing interactive decision-making systems that not only produce trustworthy results, but also provide post-hoc interpretability to facilitate interactions between model developers, users, and other stakeholders is of significant importance.

Miller (Miller, 2019) provided a detailed view about how Explainable AI can benefit from human explanation ability. Based on an extensive exploration into research on philosophy and social psychology, Miller investigated how people define, generate, choose, evaluate, and provide explanations, which can be applied by AI researchers. Although adopting the human explanation model into Explainable AI is challenging, cooperation of researchers in philosophy, psychology, and cognitive science with AI developers can pave the way for improvement in Explainable AI research.

Varona and Suarez (Varona & Suárez, 2022) proposed a review of previous research to identify discrimination, bias, fairness, and trustworthiness in the context of the social impact of AI. It was indicated that bias and discrimination are interdependent and are the cause and effect of each other. These two variables are not only dependent on algorithms and models but are also linked to data gathering, data cleaning, and data processing. Fairness refers to the unbiased outcome of AI systems. Also, authors expressed that Trustworthy AI should be built upon transparency, security, project governance, and bias management (Varona & Suárez, 2022).

Mostafazadeh Davani et al. (Davani et al., 2024) investigated the challenges of universal offensiveness standards by analyzing how cultural frameworks and individual moral foundations shaped perceptions of offensive language. In a large-scale cross-cultural experiment with 4,309 participants from 21 countries, the authors found significant variation in offensiveness judgments, where individual moral values (particularly Care and Purity) were stronger predictors than cultural background. The study revealed a critical mismatch between this moral-cultural complexity and existing AI content moderation systems, which treated offensiveness as a culture-agnostic classification task. By quantifying how moral priorities influenced sensitivity to offensive speech, the research provided empirical support for culturally adaptive moderation models to reduce systemic bias in automated systems.

Kennedy et al. (Kennedy et al., 2024) conducted research to interview 112 UK residents about how public services use data. The authors found that participants often intertwined fairness and equity in their own reasoning, linking fairness not just to equal treatment but to addressing structural disadvantages. Participants viewed fair data use as inclusive and responsive to structural disadvantages, prioritizing equitable access over equal treatment. Even without using the term "equity," their concerns reflected social justice values. The study highlights how public understandings of fairness imply solidarity and challenge technical distinctions between fairness and equity.

*Biometric and Cybersecurity:* In the realm of cybersecurity, XAI is necessary to comprehend how a ML model arrives to the decisions. Mahdavifar and Ghorbani (Mahdavifar & Ghorbani, 2020) focused on cyberattack detection through an expert system by applying trained neural network rules to unknown data. The authors of (Paredes et al., 2021) emphasized the necessity for explainability in AI-based cybersecurity and proposed a framework for the development of transparent cybersecurity software. A robust Intrusion Detection System (IDS) model was introduced by Choubisa et al. (Choubisa et al., 2022) that employs a random selection algorithm for feature selection and utilizes a Random Forest (RF) classifier. The model aims to enhance robustness in the classification of attacks.

The investigation and assessment of biometric data for business objectives is a growing area of interest for practitioners and researchers. Yang et al. (Yang et al., 2019) provided a comprehensive review of the latest developments in fingerprint-based biometrics, emphasizing security and recognition accuracy. Dhoot A. et al. (Dhoot et al., 2020) highlighted the pressing need for cybersecurity in the age of data-centric services and propose a specific model for securing transactions in the banking system through biometric impressions and digital signatures.

The research by De Keyser et al. (De Keyser et al., 2021) discussed the opportunities and challenges associated with biometric data in business, including privacy, security, and potential biases. The comprehension

and mitigation of biases have become paramount in the rapidly changing field of AI integration. Gavrilova (Gavrilova, 2023) addressed trustworthy decision-making, privacy considerations, visual knowledge discovery, and bias mitigation strategies in her discussion of Responsible AI from both a scientific and societal perspective. It provides a broad overview of bias and ethics consideration in AI-based information security applications and makes concrete recommendations for mitigating bias and reducing unfairness in AI-made decisions.

Kieslich and Lünich (Kieslich & Lünich, 2024) analyzed German public attitudes toward regulating AI-driven remote biometric identification (RBI), comparing preferences across use cases (real-time vs. post hoc application; criminal investigations vs. event security). Employing survey methodology, they demonstrated that low trust in AI or law enforcement correlates with stronger support for RBI bans, while perceived discrimination risks amplify demands for strict oversight (e.g., mandatory audits or public system registries). Notably, regulatory preferences showed minimal variation between real-time and retrospective RBI applications, suggesting pervasive public skepticism transcending contextual distinctions. The study highlights a critical gap between emergent RBI deployments and civic expectations, advocating for participatory governance models. Awumey et al. (Awumey et al., 2024) proposed a systematic review to analyze AI-powered biometric monitoring technologies used by employers—such as emotion-recognition cameras, wearable sensors, facial recognition, and physiological tracking—across their entire lifecycle: development, deployment, and usage. It finds that these systems expose workers to a spectrum of socio-technical harms including invasion of privacy, erosion of autonomy and dignity, emotional labor to suppress visible cues, biased assessments, and stress from constant surveillance . The paper highlights that harms accumulate across stages: early design flaws can ripple into deployment and ongoing use, suggesting the need for holistic, worker-centered evaluation frameworks and robust regulatory oversight to anticipate and mitigate cascading negative impacts.

*Education:* The education sector is being reshaped by introducing AI-driven decision-making. However, AI applications in education raise ethical concerns, emphasizing the need to address biases, fairness, and technical limitations. Hannan and Liu (Hannan & Liu, 2023) categorized AI applications into learning experiences, enrollment management, and student support. Their study focused on strategic decisions in areas like admission and advising to enhance the student experience in higher education institutions (HEIs) of USA. They showed how HEIs is biased through successful applications of AI technologies in three main areas of college operation: student learning experience; student support; and enrollment management.

Another research conducted by Gillani, Nabeel, et al. (Gillani et al., 2023) centers on the necessity for clarification regarding the various approaches, potential drawbacks, and capabilities that fall within the broad category of AI in education, emphasizing Intelligent Tutoring Systems (ITS), automated assessment and feedback, coaching and counseling, and large school systems-level processes. The discussion extends to the many ethical issues surrounding the use of AI in education, such as how to distribute benefits across various student groups fairly and potential risks or drawbacks of deploying this technology.

Dotan et al. (Dotan et al., 2024) developed a "Points to Consider" framework for responsible adoption of generative AI in higher education, grounded in a semester-long collaborative process at the University of Pittsburgh involving focus groups, iterative discussions, and surveys. Researchers identified key ethical and pedagogical concerns including academic integrity, equity in access, and the need for AI literacy development. Faculty emphasized the importance of maintaining human oversight in grading and feedback systems while recognizing AI's potential as a teaching aid. The resulting framework provided practical guidance for institutions navigating AI integration, balancing innovation with protection of educational values. The work highlighted discipline-specific considerations and recommended ongoing faculty training to ensure responsible implementation of emerging technologies.

*Social Media:* Anzum et al. (Anzum et al., 2022) presented a systematic review of the limitations, fairness, and biases introduced in online social media data mining and the AI model development workflow. The authors determined various factors that can introduce bias in different AI model development phases including data collection, data annotation, model training, and prediction. Therefore, this paper presented strategies to mitigate bias that involve adding datasheets for datasets, qualitative data analysis, and introducing diversity in the dataset (Anzum et al., 2022).

Trattner et al. (Trattner et al., 2022) provided a detailed overview of Responsible AI-based media technology and challenges for society and media industry presented by the rapid advancements in media technology. They introduced five research areas in responsible media technology the should be taken seriously, including understanding behavior and experiences of audiences, user modeling, personalization, and engagement, analyzing and producing media content, interaction and accessibility through media content, and Natural Language Processing (NLP) technologies in media sector.

In the evolving landscape of natural language processing (NLP), the discovery of gender biases in word embeddings and sentence encoders has sparked ethical concerns. As NLP technologies shape various aspects of our digital interactions, these biases not only mirror societal prejudices but also hold the potential to perpetuate and exacerbate existing inequalities. Caliskan et al. (Caliskan et al., 2017) developed the Word-Embedding Association Test (WEAT), similar to the Implicit Association Test (IAT), which showed the existence of gender bias in Word2Vec and GloVe word embeddings widely used in natural language processing (NLP). The proposed model was tested on large datasets containing words related to the labor force and occupation.

To mitigate gender bias from NLP tools in co-referencing, Zhao et al. (Zhao et al., 2018) proposed a data-augmentation approach combined with the existing word-embedding debiasing techniques. The proposed approach was tested on a benchmark dataset, WinoBias, where the experimental results demonstrated that gender biases were removed. Another gender bias mitigation strategy was proposed by Garimella et al. (Garimella et al., 2019) where the researchers annotated a standard parts-of-speech (POS) tagging and dependency parsing dataset with gender information. May et al. (May et al., 2019) extended WEAT and proposed Sentence Encoder Association Test (SEAT) for testing human-like implicit biases based on gender, race, and others in seven state-of-the-art sentence encoders, including ELMo and BERT.

Table 2: Key recent developments in Trustworthy and Responsible AI across various research domains

| Reference | Domain | Methods and Dataset |
|---|---|---|
| (Challen et al., 2019) | Healthcare | Clinical safety framework. |
| (Trocin et al., 2021) | | Responsible AI in digital health. |
| (Vokinger et al., 2021) | | Bias mitigation strategies across different steps of ML-based system development. |
| (Badal et al., 2023) | | Responsible AI framework and guiding principles (breast cancer research). |
| (Mittermaier et al., 2023) | | Bias in surgical AI. |
| (Giovanola & Tiribelli, 2023) | | Redefining ML-based healthcare concepts to foster social justice, and promote equity. |
| (Ueda et al., 2024) | | Fairness and sources of bias in healthcare. |
| (Hasani et al., 2022) | Medical Imaging | Exploring medical, ethical, and societal dimensions of AI in diagnosing rare diseases. |
| (Bernhardt et al., 2022) | | Providing argument on the importance of dataset bias for underdiagnosis. |
| (Stanley et al., 2023) | | Proposing a framework to assess bias in medical imaging. **Dataset:** SRI24 atlas (Rohlfing et al., 2010). |
| (Jones et al., 2023) | | Providing causal perspective on data bias for medical imaging. |
| (Xu et al., 2024) | | Providing a systematic review on fairness challenges in DL-based medical image analysis. |
| (Mercado et al., 2016) | Human-Robot Interaction | Participants collaborated with an intelligent agent to plan actions for unmanned vehicles. **Dataset:** Online survery data. |

Table 2 – *Continued from previous page*

| Reference | Domain | Methods and Dataset |
|---|---|---|
| (Hayes & Shah, 2017) | | Framed explanations using minimal set cover and Boolean logic for concise, human-friendly understanding. |
| (Cantucci & Falcone, 2020) | | Proposed an explainable HRI framework to enhance trustworthiness. |
| (Chi & Malle, 2023) | | Demonstrated dynamic trust updating when users teach a robot across sequential interactions. **Dataset:** Smartphone-based simulated teaching experiment with 220 participants over 15 trials. |
| (Kopp, 2024) | | Conceptual analysis framing trust and distrust as coexisting, multidimensional phenomena in workplace HRI. |
| (Green & Iqbal, 2025) | | Developed an objective, multimodal model of human trust using physiological signals, gaze, and facial expressions in real-time HRI. **Dataset:** In-person supervisory interaction dataset capturing BVP, EDA, skin temperature, gaze and facial cues. |
| (Koo et al., 2015) | Autonomous Vehicles | Presenting systematic regulations for ensuring the explainability of autonomous systems. **Dataset:** Qualitative user-study. |
| (Soares et al., 2019) | | Proposing a self-organizing neuro-fuzzy approach for AVs to prioritize interpretability. **Dataset:** Ford Motor dataset (Nageshrao et al., 2019). |
| (Zhang et al., 2020) | | Investigating factors that influence users' trust in autonomous driving. **Dataset:** Qualitative user-study. |
| (Yu et al., 2020) | | Publishing a diverse dataset to mitigate bias in AVs. **Dataset:** "BDD100K" dataset. |
| (Katare et al., 2022) | | Addressing biases in ML inference, emphasizing data distribution's role to improve generalization. **Dataset:** Biased-car (Madan et al., 2021) and nuScenes (Caesar et al., 2020). |
| (Asha & Sharlin, 2023) | | Demonstrating the downsides of excessive explanations in autonomous systems. **Dataset:** Qualitative user-study. |
| (Lim & Taeihagh, 2019) | | Explored ethical and technical concerns in AV algorithmic decision-making, highlighting safety risks, bias, perverse incentives, and liability gaps in smart city contexts. |
| (Ding et al., 2024) | | Demonstrated that criticality-adaptive displays, tuned by user trust propensity, age, and context, improve situational awareness and trust in automated driving. |
| (Ling et al., 2024) | | Integrated uncertainty-aware explanations into perception models, showing that communicating prediction confidence supports better trust calibration and decision alignment. |
| (Mahdavifar & Ghorbani, 2020) | Biometric and Cybersecurity | Formed DeNNeS to address the lack of explainability in DL models. **Dataset:** Phishing websites and malware. |

Table 2 – *Continued from previous page*

| Reference | Domain | Methods and Dataset |
|---|---|---|
| (Paredes et al., 2021) | | Proposing a roadmap for transparent AI-based cyberse-curity.
**Dataset:** National Vulnerability Database. |
| (Choubisa et al., 2022) | | Presenting Random Forest to distinguish DoS, Remote to Local attack, Probe, and U2R.
**Dataset:** Kaggle dataset "NSL-KDD". |
| (Dhoot et al., 2020) | | Building architecture to prioritize optimal user interaction with information resources.
**Dataset:** "FVC2002DB1B". |
| (De Keyser et al., 2021) | | Guidelines for responsible biometric empirical studies and multidisciplinary approaches. |
| (Gavrilova, 2023) | | Proposing bias mitigation strategies in AI for biometrics, social media, and cybersecurity.
**Dataset:** Biometric datasets. |
| (Kieslich & Lünich, 2024) | | Advocating participatory governance for regulating AI-driven remote biometric identification. |
| (Awumey et al., 2024) | | Proposing a lifecycle-based, worker-centered evaluation framework to address socio-technical harms of AI-powered biometric monitoring in the workplace. |
| (Hannan & Liu, 2023) | Education | Integrating AI in various aspects of higher education to enhance the student experience. |
| (Gillani et al., 2023) | | Emphasizing benefits and limitations, underscoring the importance of ethical awareness. |
| (Dotan et al., 2024) | | Developing a collaborative, faculty-informed framework for responsibly integrating generative AI in higher education. |
| (Anzum et al., 2022) | Social Media | Review on data checking, model development, and bias mitigation strategies. |
| (Caliskan et al., 2017) | | Investigating gender bias in word embeddings.
**Dataset:** "Common Crawl" corpus. |
| (Zhao et al., 2018) | | Mitigating gender bias in co-reference systems through rule-based gender-swapping.
**Dataset:** "WinoBias". |
| (Garimella et al., 2019) | | Proposing a data annotation strategy to mitigate gender bias in text data.
**Dataset:** Wall street journal. |

# 6  Open Challenges

The rapid integration of AI systems across fields, from healthcare to finance, heralds a transformative era in decision-making. Yet, these advancements are accompanied by multifaceted challenges that raise questions about the trustworthiness and widespread adoption of such technologies.

*Ethical Development and Deployment of AI:* A foundational challenge, regardless of domain, is the ethical development and application of AI. As AI systems permeate diverse sectors, striking a balance between model accuracy and ethical principles, such as fairness, transparency, and interpretability, becomes an essential concern. While precise outcomes are pivotal for AI's efficacy, they must align with ethical principles to ensure that all individuals and communities are treated equitably.

*Data Privacy:* Data privacy is another omnipresent challenge. Particularly in sensitive areas like medicine or banking, privacy considerations often lead to the omission of critical attributes from datasets, such as age, sex/gender, race/ethnicity, or socioeconomic status. Such omissions can inadvertently introduce biases, complicating the process of bias detection and mitigation. Moreover, as AI solutions are globalized, harmonizing their functions with diverse data protection laws and regulations is paramount.

*Regulation and Policy Development:* A significant ongoing challenge involves developing regulations and policies for AI that are sufficiently robust to keep up with the rapid advancements in the field. These regulations should not only foster innovation but also safeguard public interests and address societal concerns. This balance is crucial to ensure that AI development progresses in a responsible and beneficial manner for all stakeholders.

*Long-term AI Safety and Control:* Advanced AI systems, due to their complexity, may evolve in unpredictable manners, making it difficult to anticipate their long-term effects. As these systems become increasingly sophisticated, ensuring human control over them presents a growing challenge. Research in this field is directed towards recognizing potential risks, particularly those stemming from unintended outcomes of AI decisions. This research also involves devising strategies to ensure persistent human oversight and effectively mitigate these risks, even when AI systems may exceed human intelligence in certain domains.

*Sustainable AI Development:* Sustainable practices should be integrated across the AI lifecycle, focusing on ecological integrity and social justice. Some key challenges for sustainable AI development include its significant environmental impact, particularly its carbon footprint, high energy consumption, and the ethical dilemmas posed by resource allocation. Additionally, it is crucial to create a balance between technological innovation and environmental sustainability, and to raise awareness among all stakeholders about AI's environmental costs. Effective policies need to govern AI's environmental impact and ensure its development aligns with broader societal values and sustainability goals (Van Wynsberghe, 2021).

*Governance:* The governance of AI systems is a multifaceted challenge that operates on several levels, ranging from research groups to overarching governmental frameworks. While each level plays a critical role in ensuring Responsible AI development and deployment, the delineation of responsibilities and the activation points for different governance layers remain unclear. At the research group level, the focus is often on ethical considerations and compliance with institutional policies. Institutions should establish guidelines to balance innovation with social responsibility. However, the role of government in AI governance introduces an additional layer of complexity. Determining when and how government intervention should occur, and the form it should take, is a subject of ongoing debate. This debate involves considerations that can be political in nature, although this aspect falls outside the scope of this paper. There is an urgent need for more discourse and clarity around the mechanisms and principles guiding AI governance across these various levels, to ensure that AI development is both ethically sound and socially beneficial.

*Data Collection:* Translating AI models for diverse and larger populations introduces a host of challenges. These encompass biases, poor generalization, inconsistent findings, and in some contexts, issues of limited clinical relevance. Such hurdles can restrict the broader adoption and utility of AI tools. To mitigate these challenges, it is vital to ensure diverse geographical representation in data collection. A predominant focus on data from high-income countries, such as Europe and North America, can lead to imbalances in race/ethnicity representation. This, in turn, can result in AI systems performing unequally across different demographic groups, leading to potential unfairness. The omission of data from underrepresented groups exacerbates this imbalance. Hence, by actively including data from marginalized communities and minority populations, the risk of selection bias can be significantly reduced. Nevertheless, we must recognize that achieving diverse data collection is itself a significant challenge. Often, researchers rely on readily available datasets rather than procuring new data, which can perpetuate existing gaps. The scarcity of data pertaining to underrepresented groups may stem from various factors. These include limited access to the services from which data are collected or a lower propensity of these groups to utilize such services. Additionally, the methods employed in data acquisition may not be adequately designed to capture

information from diverse individuals effectively. Addressing these systemic issues is crucial for ensuring equitable representation in AI-driven research and applications.

*Generalization:* As discussed in various sections of this paper, the concept of generalizability is pivotal in AI development. Ideally, one might envision a universal model that performs optimally across all demographics. At the other end of the spectrum are models tailored for individual users. In practice, AI models typically fall between these extremes. The necessity of developing either region-specific or ethnicity-specific models, or those that uniformly cater to diverse demographics, presents a significant challenge. This challenge must be addressed on a case-by-case basis for each application. Regardless of the approach, a fundamental practice should be the transparent communication of a model's limitations and target demographic. It is essential to clearly outline which groups are likely to benefit from the model and acknowledge those for whom the model has not been evaluated or may underperform. Such transparency is vital to prevent misuse and ensure responsible application of AI technologies.

*Differences in Stakeholders perspectives:* For example in recidivism, the police/decision maker cares more about less False Negatives (predictive value) whereas defendants, particularly those wrongly classified as future criminals, want False Positive classifications to be less. The society might also focus more on if the selected set is demographically balanced. So, different metrics matter to different stakeholders and there could be no "right" definition for group fairness.

Explainable AI Limitation: Many XAI systems prioritize technical clarity but overlook end-user needs. Users in different roles—clinicians, regulators, or laypeople—require explanations tailored to their specific tasks and contexts. Without this customization, explanations may fail to support actionable understanding, leading users to trust systems blindly or miss critical limitations. Human-centered XAI faces its own challenges: there is no universal standard to assess explanation quality for human audiences, and poorly designed explanations can overwhelm users or distort their judgment. Faithfully representing complex black-box models (e.g., large language models (LLMs)) is inherently difficult, and striking the right balance between fidelity and intelligibility remains a hurdle—over-simplified explanations risk misleading users, while overly technical ones may confuse them. Additionally, explanations can inadvertently amplify biases or mask discrimination in AI systems. Practical deployment introduces further obstacles; many XAI methods succeed in labs but falter in production due to latency, usability, or integration issues. For instance, generating real-time explanations for LLMs often proves computationally prohibitive (Kong et al., 2024).

*Technical Limitations of AI Systems:* Technical limitations in AI systems present significant challenges across various domains. The need for Trustworthy and Responsible AI-driven decision-making is paramount. A key issue is performance limitations of AI systems and the potential impact of inaccurate predictions, which can have far-reaching consequences. To ensure reliability, AI systems typically require extensive data collection over long periods to adequately represent diverse outcomes. This necessity often leads to a measured and cautious approach in integrating AI into critical decision processes. The "information barrier" adds to these challenges. Many AI systems operate as "black boxes," with opaque decision-making processes. This lack of transparency makes it difficult to understand the logic behind AI predictions, leading to skepticism about their trustworthiness and reliability, particularly in sensitive applications like clinical trials.

*Healthcare Perspective:* Advancements in AI have paved the way for the application of AI in healthcare, however, this trend has raised significant ethical issues that should be addressed. Data bias and algorithmic bias, lack of transparency, and technical limitations preclude harnessing the full potential of AI for clinical practices. Biased AI algorithms for healthcare applications lead to inaccurate predictions/recommendations, harmful outcomes, or discriminatory practices. Patients may receive incorrect diagnoses or delayed treatment, worsening health conditions and increasing morbidity. The cost of treating advanced or unmanageable diseases can be significantly higher than early intervention. A Trustworthy AI framework, by combining clinical information, demographics, and imaging, will redefine the existing AI-based methods for healthcare applications. Trustworthy AI enhances reliability, reduces errors, applies consistency across

populations, and mitigates potential harm to participants, which in turn can exert a positive impact on healthcare costs.

*Accountability Frameworks in Healthcare AI:* A critical challenge lies in establishing clear accountability structures for AI-driven healthcare systems. While AI offers diagnostic support, determining liability for errors requires delineating responsibilities among physicians (who must critically evaluate AI outputs), developers (who ensure algorithmic fairness and transparency), and institutions (who govern clinical integration). Current frameworks often lack mechanisms to address harm from biased predictions or misdiagnoses, creating legal and ethical uncertainties. Robust accountability protocols must balance innovation with patient safety, ensuring all stakeholders understand their roles in maintaining trustworthy AI systems while allowing for continuous improvement through feedback loops. This challenge intersects with broader needs for regulation and explainability (Ueda et al., 2024).

*Job Security:* The advent of AI systems brings considerable advantages across multiple sectors, enhancing efficiency and accuracy in various tasks. However, this technological advancement also raises widespread concerns regarding job security. AI applications, which are designed to optimize operations, often gradually undertake roles that have been traditionally performed by human employees in areas such as customer service, manufacturing, healthcare, and even creative industries. The challenge, therefore, lies in developing AI technologies that augment and enhance human capabilities, rather than replacing them, ensuring a balance between technological progress and job preservation.

In essence, as we traverse this AI-augmented landscape, it is imperative to confront both general and domain-specific challenges to truly harness the transformative power of AI.

## 7 Guidelines and Recommendations

In this section, we elaborate on a set of guidelines aimed at fostering the development of Trustworthy AI, by focusing on best practices that ensure the reliability and unbiased nature of AI models.

*Adaptive AI Models and Continuous Learning:* As the technology landscape advances, there is an undeniable need for AI models that can adapt and learn continuously. While the infusion of new data to these models is integral, it presents a challenge: even unbiased data can inadvertently introduce or amplify system biases. This necessitates the consistent application of bias detection and mitigation mechanisms to ensure system integrity.

*Human-Centered Design Approach:* Emphasizing a human-centered methodology is pivotal, especially in sectors like medicine where decision-making is critical. Such an approach requires the integration of human insight across various AI development phases—from crafting algorithms to pinpointing and rectifying errors. This symbiotic relationship between human cognitive ability and AI's computational strength optimizes decision-making processes, ensuring both the reliability and trustworthiness of the system.

*Multi-Metric Assessment and Direct Data Examination:* When designing AI systems, it is essential to adhere to the best practices typically associated with software systems. However, ML introduces its own set of unique considerations. Emphasizing the identification of multiple metrics for training and monitoring is pivotal. Using a variety of metrics allows for objective evaluations, enabling a comprehensive assessment and improvement of model performance. Moreover, whenever feasible, a direct examination of the raw data is imperative. The nature of this examination varies depending on the data's type and volume, ranging from visual inspections and Exploratory Data Analysis to statistical testing. Such scrutiny can preemptively detect and address biases in the dataset before training begins.

*Cost-effectiveness:* More comprehensive research is essential to evaluate the cost-effectiveness of AI systems when applied for varying tasks. AI tools should provide acceptable services and outcomes for the same cost as existing tools or more acceptable outcomes for less cost. Indeed, the cost of data gathering, developing, updating, implementing, and maintaining AI systems should be considered through a comprehensive and uniform measurement system.

*Understanding Limitations and Continuous Monitoring:* A well-rounded AI system acknowledges its limitations. Through comprehensive testing and performance evaluations, developers can gauge the boundaries of datasets and models. Post-deployment, the focus shifts to perpetual monitoring and regular updates. This continuous cycle not only ascertains bias minimization but also guarantees optimal real-world system performance.

*Robust Data Collection and Governance:* The foundation of any AI system is its data. Aggregating diverse and representative datasets, coupled with generative techniques to augment data diversity, becomes paramount. Expanding dataset diversity—by incorporating data from varied geographical regions, demographics, and especially from underrepresented communities—enhances model generalizability, thereby reducing potential biases.

*Providing datasheets for datasets:* Data is considered to be a foundation for training and evaluating the model, thus, its characteristics exert a considerable influence on the model's performance. Unintended biases and misalignment in datasets between training and deployment bring about negative effects on the model performance. These problems necessitate creating datasheets for datasets to improve transparency and mitigate biases in ML models. These datasheets enable users to gather sufficient information about datasets and make informed decision about the most appropriate one for their specific task. Some of the main information that should be included in datasheets are as follows: purpose and origin of the dataset; critical details about the instances; collection process; preprocessing/cleaning/labeling stages; previous and future applications of dataset and its limitations; distribution methods; and ongoing support and maintenance of the dataset. This information enables users to understand the context and biases introduced during data creation and contribute to the enhanced transparency of datasets (Gebru et al., 2021).

*Collaboration of all stakeholders:* For an AI system to be sustainable, active collaboration between all stakeholders is significant. For example, stakeholders directly or indirectly affected by diseases, including patients, healthcare providers, payers, and regulatory agencies should be involved in the development, design, and deployment of AI systems. This, in turn, can improve the trustworthiness of AI systems for clinical applications. Continuous education of patients and medical professionals regarding advancements in AI-based diagnostic systems, together with their opportunities and challenges can build trust in the application of these systems for medicine.

*Educational Practices:* The importance of fairness and reliability is not limited to the healthcare system. The fundamental approach is to provide students in different fields, ranging from Finance and Law to Computer Science and Software Engineering, with educational practices and detailed knowledge about fairness, reliability, and responsibility across their areas of specialization. For example, in Computer Science and Software Engineering, it is essential to include concepts related to Fairness in AI and Responsible AI in educational standards. Continuous, standardized, and in-depth education can foster deep and critical thinking toward fairness across different sections.

*Social and environmental well-being:* When designing, developing, and deploying AI systems for different purposes, it is essential to ensure that the well-being of people and the environment is at the forefront of Trustworthy AI considerations (Kaur et al., 2022). For example, developing DL models causes environmental costs due to the carbon footprints of generating modern tensor processing hardware as well as the energy required to power the hardware for protracted model training. Thus, designing and developing a framework for real-time energy consumption and carbon emissions tracking is recommended to mitigate environmental hazards caused by AI systems. Through such frameworks, carbon footprint could be estimated by specifying hardware type, time span, and cloud provider used for designing and implementing an AI system. Developing effective tools to enable models to stop training when it exceeds a responsible energy consumption or carbon emission is fundamental (Van Wynsberghe, 2021).

*Human-Centered Explainable AI*: Explanations must adapt to user roles, domains, and real-time needs, but most XAI is static. To overcome this, human-centered XAI calls for designing interactive and adaptive

explanation systems that tailor content, complexity, and presentation based on the specific stakeholder's background, goals, and decision context. For example, end users need accessible and actionable insights, while data scientists require technical transparency for model debugging, and regulators seek clarity on compliance and fairness. These diverse needs necessitate XAI systems that support user-driven exploration, allow follow-up questions, and dynamically adjust explanation granularity, enabling more meaningful, context-aware engagement and improving trust, usability, and accountability in AI systems (Kong et al., 2024).

*Industry Perspective:* Adhering to developing Trustworthy AI systems not only in academia but also in the industry is considered imperative. Similar to the academic context, AI systems in industry should comply with regulations to be lawful, ethical, and robust. An appropriate legal framework is essential to obligate industries to pursue public policy, engage in a fair market, and pave the way for economic benefits while maximizing protection for citizens. Collaborating with stakeholders to engage them in the development and deployment of AI systems in the industry and collecting their feedback can fulfill the ethical aspect. To ensure robustness, identifying risk and safety factors relevant to the design and adoption of AI systems together with the compliance of these factors with standards should be considered (Radclyffe et al., 2023).
The European Commission's High-level Expert Group on AI established an Assessment List for Trustworthy Artificial Intelligence (ALTAI) tool to provide organizations with a web-based self-assessment checklist to ensure that guidelines are met by users. Prior to the application of the ALTAI tool, some factors are considered to ensure the Fundamental Rights Impact Assessment (FRIA) in the context of the design and implementation of a new legislative instrument. These factors include negative discrimination against people regarding protected attributes such as gender and ethnicity, child protection laws to prevent any harm to child users, data protection for privacy concerns, and fundamental freedom of citizens to prevent limits and impacts on their freedom (Radclyffe et al., 2023).
The European Commission's High-level Expert Group on AI introduced ethical guidelines for Trustworthy AI, which are human agency and oversight, technical robustness and safety, privacy and data governance, transparency, diversity, non-discrimination, and fairness, societal and environmental well-being, and accountability (Commission, 2022). The ALTAI tool follows these seven principles when considering trustworthiness in the industry. Human agency and oversight deals with considering human contribution toward the activity of the AI system and providing feedback and correction to improve its performance. Technical robustness and safety can be fulfilled by considering risks, risk metrics, and risk levels of AI systems in specific use cases and monitoring these systems to detect and minimize potential harm. For some AI systems to operate, a level of personal data and personally identifiable information is required. Thus, considering privacy and data governance in the design process of AI systems is crucial. Transparency refers to having control over the data and model used for a specific decision-making process in AI systems, evaluating the extent to which the outcome of the AI system is explained to the users, and assessing the users' understanding of the decisions made by these systems. To ensure diversity, non-discrimination, and fairness in the industry, AI systems should be user-centric and designed to allow all people to adopt them. Also, the diversity and repetitiveness of end users and subjects in the data should be taken seriously. Societal and environmental well-being is essential to detect potential negative impacts of AI systems on the environment and society. Accountability should be considered to monitor and assess the AI system to ensure compliance with ethics and key requirements for Trustworthy AI in the industry (Commission, 2022).
One example of applying the "Framework for Trustworthy AI", developed by the independent High-Level Expert Group of AI, is a novel process called Z-Inspect. This process is proposed to assess the trustworthiness of AI models based on the mentioned ethical guidelines for Trustworthy AI and it is applicable to various domains, such as business, healthcare, and public sectors, among others (Zicari et al., 2021).

## 8 Conclusion and Final Remarks

In this study, an attempt is made to define AI principles and discuss challenges and risks in this context. Here, we highlighted Trustworthy AI, Responsible AI, and Fairness in AI as foundational AI principles. Further, some challenges and risks introduced by the advent of AI are explored, namely, personal privacy, disinformation, data limitation, explainablility and transparency, and biases. We also explored Fairness in

AI, identifying and categorizing various sources and types of bias and discussed several bias detection and mitigation techniques, as well as bias evaluation metrics.

We highlighted open challenges observed in a variety across different domains, from healthcare to finance. Lack of standardization in ethical development and Trustworthy AI is considered one of the main challenges. Comprehensive and integrated ethical principle guidelines should be developed to ensure that AI systems meet requirements, including, privacy and data protection, diversity, fairness, accountability, transparency, human well-being and human rights, and sustainable development. However, developing international standards for Trustworthy AI presents other challenges due to the slow and varying response of national governments to AI regulation, or even political concerns.

We discussed some guidelines and recommendations to foster the development of Trustworthy AI. Developing adaptive models that can learn, detect bias, and mitigate them continuously is essential in this context. Considering a human-centric design approach, continuous monitoring, and understanding of the limitations of AI systems are critical to ensure the reliability and trustworthiness of systems. Applying various model evaluation metrics and data examination approaches can improve the performance of the model and provide insight into its generalization capability. To improve transparency and reduce biases in AI systems, data creators should adopt standardized datasheets to document key aspects of the data's lifecycle. These datasheets should include information on the origin of the dataset, collection process, demographic representation, data processing stages, potential risks and biases, limitations, and privacy considerations. This increased transparency reduces the risk of deploying biased models and promotes more responsible and fair use of data in sensitive domains such as healthcare, criminal justice, and finance. Finally, the participation of all stakeholders is recommended during the development, design, and deployment of AI systems to allow for a more detailed insight into various aspects of trustworthiness.

### Acknowledgments

The authors acknowledge NSERC and Alberta Innovates for financial support, and the Office of the Vice-President for Research at the University of Calgary and the Transdisciplinary Initiatives for support and the excellent idea stemmed.

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
