# OpenReview forum: "Trustworthy and Responsible AI for Human-Centric Autonomous Decision-Making Systems"
_TMLR — Accepted by TMLR_

### Review · Reviewer_hJaL · 2025-06-26

**Summary Of Contributions:**

This paper presents a comprehensive interdisciplinary review of trustworthy and responsible AI principles and their practical implementation. The authors make several key contributions:

* **Conceptual Framework:** The paper establishes a clear definition of trustworthy AI based on Lyu et al. (2021) and articulates five foundational principles: beneficence, non-maleficence, autonomy, justice, and explicability. It systematically defines and examines responsibility, explainability, and fairness in AI through existing literature.
* **Governance and Implementation:** The authors provide a structured approach to AI governance, covering ethical frameworks, data security protocols for user consent and privacy, and regulatory compliance mechanisms that translate AI regulations into actionable development policies. They emphasize reproducibility, reliability, and effective communication as essential components of explainability.
* **Bias Analysis and Mitigation:** A central contribution is the comprehensive analysis of bias in AI systems, including taxonomy of bias types and sources presented through a feedback loop model spanning data, algorithms, and users. The paper proposes systematic interventions across the algorithm lifecycle—data collection/preprocessing, algorithm design, and output evaluation—with existing bias evaluation metrics summarized in Table 1.
* **Domain-Specific Guidelines:** The authors provide tailored trustworthy AI recommendations for eight human-centric application domains: healthcare, medical imaging, human-robot interaction, autonomous vehicles, philosophy and social sciences, biometrics and cybersecurity, education, and social media, with comprehensive guidelines presented in Table 2.
* **Practical Implementation:** The paper addresses current challenges in AI-driven decision-making systems (ethics, privacy, regulation, employment) and proposes actionable best practices emphasizing adaptive systems, human oversight, holistic evaluation, multi-stakeholder collaboration, and diverse, well-documented datasets to inform users of potential biases.

**Audience:**

Yes

**Claims And Evidence:**

Yes

**Requested Changes:**

In section 2, I would add to an environmental consideration to the non-maleficience aspects of trusworthy AI (3rd paragraph). The sentence "In autonomy, the aim is to trade-off decisions made by human against decisions made by machine to ensure integrity and reliability of AI systems" is unclear and should be rephrased. In the paragraph introducing responsible AI, can you please provide a clear definition that removes the ambiguity between responsible AI and trustworthy AI. In the 6th paragraph, there is a missing citation when mentioning Fujisto's experiment. The last paragraph briefly recommends AI deployment that respects the environment, but it is not mentioned elsewhere in the section and should be introduced as it is a critical risk that AI development raises.

In section 3, can you please define in a clear way what you refer to as reproducibility ? The absence of the definition makes it seem that it is confused with explainability.

In section 4, the third paragraph mentions a really interesting idea of positive discrimination in the development of trustworthy systems, but the paragraph is a bit confusing. I believe it needs rephrasing as it is a very important idea !
In the paragraph "Algorithms to users": shouldn't popularity bias be in the section data to algorithms ? if not, can you explain why it fits better in algorithms to users ? Finally, the paragraph just before section 4.1 is not really related to the rest of the section.

In section 4.1, the definition of fairness through awareness is unclear and needs rephrasing.
In section 4.3, besides evaluating bias, what would be bias mitigation examples ? Can you include them in Figure 2 ?

**Strengths And Weaknesses:**

### Strenghts:
This is an extensive review of trusworthy and responsible AI, with a focus on bias issues that goes beyond simple algorithmic design. It provides a realistic framework of AI development from data collection to output monitoring. The reader can also see that this was a work done by a multidisciplinary team, as interesting and domain specific insights were given all across the study. Although the material is dense, the paper unified a wide range of definitions and provided a good summary of the literature, especially in Table 1 for fairness evaluation metrics and section 5 for the specific study of trusworthiness for different human centric applications.

### Weaknesses:
There is a few redundancies when introducing certain concepts, and I believe that it is mainly due to the disparity of definitions across the literature. For instance, the definition of Responsible AI in section 2 seems encapsulated by the non-maleficience and explicability aspects of trustworthy AI. Similarly, the 4th paragraph of section 3 introduces the notion of reproducibility, the definition is ambiguous and seems very analogous to explainability.
The bias mitigation discussion lacks concrete implementation guidance, proposing primarily evaluative approaches at each algorithm design stage without sufficient practical examples. While Section 4.1 mentions relevant toolkits, no tangible implementation examples are provided. Figure 2 represents a missed opportunity, presenting the bias feedback loop without accompanying mitigation strategies that would substantially enhance the paper's practical value.

---

> ### Author Response · Authors · 2025-07-21
> **Response to Reviewer hJaL’s Comments**
>
> We sincerely appreciate the time and careful consideration you have devoted to reviewing our manuscript. Below, we provide detailed point-by-point responses to your comments, with all corresponding revisions highlighted in yellow in the manuscript.
>
> **Comment:** In section 2, I would add to an environmental consideration to the non-maleficience aspects of trusworthy AI (3rd paragraph).
>
> **Response:** Thank you for your valuable suggestion. We have now included environmental considerations as part of the non-maleficence aspect of trustworthy AI in Section 2, paragraph 3.
>
> ---
>
> **Comment:** The sentence "In autonomy, the aim is to trade-off decisions made by human against decisions made by machine to ensure integrity and reliability of AI systems" is unclear and should be rephrased.
>
> **Response:** Thank you for your feedback. We have rephrased the sentence for clarity.
>
> ---
>
> **Comment:** In the paragraph introducing responsible AI, can you please provide a clear definition that removes the ambiguity between responsible AI and trustworthy AI.
>
> **Response:** Thank you for your suggestion. We have revised the paragraph to clarify the distinction between responsible AI and trustworthy AI.
>
> ---
>
> **Comment:** In the 6th paragraph, there is a missing citation when mentioning Fujisto's experiment.
>
> **Response:** Thank you for catching this oversight. We have added the appropriate citation for Fujisto's experiment in the 6th paragraph.
>
> ---
>
> **Comment:** The last paragraph briefly recommends AI deployment that respects the environment, but it is not mentioned elsewhere in the section and should be introduced as it is a critical risk that AI development raises.
>
> **Response:** Thank you for highlighting this important point. As suggested, we have expanded Section 2 to include a dedicated discussion on the environmental risks of AI development.
>
> ---
>
> **Comment:** In section 3, can you please define in a clear way what you refer to as reproducibility? The absence of the definition makes it seem that it is confused with explainability.
>
> **Response:** We appreciate this important clarification. In Section 3, we now explicitly define reproducibility as "the ability to consistently replicate an AI system's results using the same methods, data, and computational environment", distinguishing it from explainability, which focuses on interpreting how decisions are made.
>
> ---
>
> **Comment:** In section 4, the third paragraph mentions a really interesting idea of positive discrimination in the development of trustworthy systems, but the paragraph is a bit confusing. I believe it needs rephrasing as it is a very important idea!
>
> **Response:** We sincerely appreciate your valuable feedback about the discussion of positive discrimination in trustworthy AI systems (Section 4). We have carefully reworked the paragraph for more clarification.
>
> ---
>
> **Comment:** In the paragraph "Algorithms to users": shouldn't popularity bias be in the section data to algorithms? if not, can you explain why it fits better in algorithms to users?
>
> **Response:** We appreciate this insightful comment. While popularity bias originates from skewed data (the "data to algorithms" phase), we position it in the "algorithms to users" section because its harmful effects, such as feedback loops and reduced diversity, occur when algorithms actively recommend and amplify popular content to users. The real-world impact occurs at this final stage, where algorithmic output shapes user experience and perpetuates inequitable exposure. We have refined the paragraph to better articulate this distinction between the origin of the bias and where its impact becomes most evident.
>
> ---
>
> **Comment:** Finally, the paragraph just before section 4.1 is not really related to the rest of the section.
>
> **Response:** Thank you for this observation. Upon review, we agree that the paragraph was unrelated to the section's focus. We have therefore removed this content to improve the paper's coherence and flow.
>
> ---
>
> **Comment:** In section 4.1, the definition of fairness through awareness is unclear and needs rephrasing.
>
> **Response:** Thank you for your feedback. We have revised the definition of "fairness through awareness" in Section 4.1 to improve clarity.
>
> ---
>
> **Comment:** In section 4.3, besides evaluating bias, what would be bias mitigation examples? Can you include them in Figure 2?
>
> **Response:** Thank you for your helpful comment. In response, we have added concrete examples of bias mitigation techniques corresponding to the three categories: pre-processing, in-processing, and post-processing. While we considered incorporating these examples directly into Figure 2, we concluded that doing so would significantly increase the visual complexity and potentially hinder readability. Instead, we have added the examples to the caption of Figure 2, where they remain easily accessible without cluttering the figure itself.

---

### Review · Reviewer_ogLD · 2025-07-09

**Summary Of Contributions:**

This paper presents a survey of many of the challenges in trustworthy and responsible AI. The paper is written for a general audience and presents many existing problems in this subfield. The paper presents open challenges and some high-level suggestions for what problems researchers should work on.

**Audience:**

No

**Broader Impact Concerns:**

This paper does not have a Broader Impact Statement section.

**Claims And Evidence:**

No

**Requested Changes:**

* Focus on a narrower topic. Pick one of the topics and go deeper into that topic. Maybe it should be AI in Healthcare or Privacy with respect to AI. I would avoid fairness and bias, since they are quite charged and there is not a good framework for working on them despite a substantial amount of work over the past decade or so.
* There is no specific evidence to support any of the claims made in the paper. Depending on the claims the authors would like to make, they should provide sound and complete evidence for those claims with data or citations.
* Write the paper for a machine learning audience. The paper is currently written more for a general audience, but it should be written for researchers who can actually do the work to fix the issues discussed in the paper. Details in this case matter a lot. There is quite a lack of detail and focus in the paper.

**Strengths And Weaknesses:**

### Strengths
* The paper is overall well-written and is concerned with an important topic in general.

### Weaknesses
* This paper is not a good fit for TMLR. It is written for a more general audience and not for the ML community. The paper also does not give actionable advice to ML researchers for how they should address the concerns in the paper.
* The paper tries to cover way too many topics at a very surface level. In doing so, the paper ends up not discussing any of the topics in sufficient detail, and thus, does not cover any of the topics well.
* The paper provides quite an out-dated perspective. For example, the paper focuses on out-dated terminology such as "Explainable AI" and completely ignores modern work on, for example, mechanistic interpretability.
* There is not any real data or results to support the claims in the paper.
* The position(s) presented in the paper are very vague and essentially useless to ML researchers even superficially familiar with these areas.

---

> ### Author Response · Authors · 2025-07-24
>
> We sincerely thank you for the time and constructive feedback you have provided in reviewing our manuscript. We fully understand and appreciate your concerns regarding the paper’s relevance and interest to the ML community. We are actively revising the manuscript to incorporate a more in-depth discussion on modern terminologies and perspectives—such as human-centered XAI, interpretability in foundation models like LLMs, and mechanistic interpretability. We are also removing outdated terminology and strengthening our arguments with additional citations, especially in Sections 6 (Open Challenges) and 7 (Guidelines and Recommendations).
>
> We would also like to mention that the reviews received from the other two reviewers were positive, emphasizing the value and relevance of this transdisciplinary contribution. We recognize that, like you, they also asked for greater clarity in the definitions and focus of the manuscript. We are actively working on addressing these concerns.
>
> We also wanted to respectfully highlight that our work has 13 citations in arXiv to date (title included here for reference), illustrating our manuscript value as a foundational reference, supporting ML researchers and domain experts align their efforts around core principles of trustworthy AI. And your feedback is helping us enhance the clarity, rigor, and relevance of the manuscript, and we are sincerely grateful for your engagement.
>
> Thank you again for your thoughtful and valuable review.
>
> Title of paper citations for reference:
>
> [1] Evaluating human-ai collaboration: A review and methodological framework
>
> [2] Achieving on-site trustworthy AI implementation in the construction industry: A framework across the AI lifecycle
>
> [3] Bridging the gap: From AI success in clinical trials to real-world healthcare implementation—A narrative review
>
> [4] Interacting with ai reasoning models: Harnessing" thoughts" for ai-driven software engineering
>
> [5] Enhancing Human-Centric Logistics Decision-Making with AI-Driven Route Optimization and Predictive Insights
>
> [6] Enhancing Trust or Fostering Misjudgment? Assessing the Impact of Emerging Geographic Information Displays on Social Media Users' Information Trust
>
> [7] Rethinking Technological Investment and Cost-Benefit: A Software Requirements Dependency Extraction Case Study
>
> [8] Easy to read, easier to write: the politics of AI in consultancy trade research
>
> [9] Artificial Intelligence in Manufacturing Industry Worker Safety: A New Paradigm for Hazard Prevention and Mitigation.
>
> [10] Perceptions of Agentic AI in Organizations: Implications for Responsible AI and ROI
>
> [11] Applying Ethics in AI-Based Research: Opportunities and Challenges
>
> [12] Investigating Sex-Related Bias in Brain MRI using Fair and Explainable Deep Learning
>
> [13] An institutional approach to values: perspectives on scientific data sharing and reuse

---

> > ### Author Response · Authors · 2025-08-03
> > **Response to Reviewer ogLD’s Comments**
> >
> > We sincerely appreciate the reviewer’s thoughtful feedback. In response, we have updated the manuscript to incorporate the suggested concepts and terminology. The newly added sections, including key conceptual updates, are highlighted in the text. Additionally, several minor revisions, including improved phrasing, added citations, and incorporation of updated terminology, were made throughout the manuscript to enhance clarity and completeness. These minor edits are not individually highlighted but collectively contribute to a more refined and comprehensive presentation.

---

> > > ### Comment · Reviewer_ogLD · 2025-08-05
> > >
> > > Thank you updating the paper and considering my feedback. I think the paper is in a much better state now.
> > >
> > > I would like to comment that I do not think that having citations is a good reason to accept a paper.
> > >
> > > Despite this, I do think the paper is acceptable now.

---

> > > > ### Author Response · Authors · 2025-08-06
> > > > **Response to Reviewer ogLD’s Comment**
> > > >
> > > > We fully agree with your point. We truly appreciate your guidance and grateful for your acceptance recommendation.

---

### Review · Reviewer_AwCm · 2025-07-20

**Summary Of Contributions:**

This paper serves as a comprehensive review of the principles underpinning trustworthy and responsible AI and related challenges and open questions. The authors being by first defining Trustworthy AI and Responsible AI and the associated fundamental principles. They also venture into a description of Explainable AI (XAI, Fairness in AI and Privacy in AI. Subsequently, the paper discusses AI governance, data security and regulatory compliance. Sections 2 and 3 are then tied together with an outline of all the foundational principles of Trustworthy AI, with reproducibility, reliability and communication stated to be foundational pillars.

Section 4 focuses on a discussion of biases and discrimination, wherein the authors make a distinction between the two. Then, they discuss the categories of bias and where it can creep into the AI development cycle – “data to algorithm”, “algorithm to user” and “user to data”. Existing literature on strategies to detect and mitigate bias in pre-processing, in-processing and post-processing, are discussed in Section 4.1. A fairly extensive of evaluation metrics for bias has also been included (Table 1).

The paper then pivots to a literature review of Trustworthy and Responsible AI in various human-centric applications. Finally, the authors close out with a delineation of open challenges in this space, and guidelines and recommendations aimed at encouraging the development of Trustworthy AI.

**Audience:**

Yes

**Broader Impact Concerns:**

I do not have any broader impact concerns with this work.

**Claims And Evidence:**

Yes

**Requested Changes:**

**Critical changes:**

1.	Pursuant to the comments from the previous section, I would certainly like to see an effort made to amend the text in Sections 2 and 3 to improve the clarity of the text. This is important as these serve as key contributions of this work.

   a.	It is not clear what the distinction between explainability and interpretability is.
2.	On Page 3, the paper names anonymization and federated learning as potential solutions to patient privacy. Both these strategies have to be shown to be ineffective in adversarial scenarios. A qualification of where these strategies can be effective is needed here. A similar claim is made in the first paragraph of Section 3.
3.	The mathematical definition of demographic parity in Table 1 appears to be incorrect.

**Suggested changes:**

1.	When discussing autonomy on Page 2, it is stated that “In autonomy, the aim is to trade-off decisions by human against decisions made my machine”. How is this autonomy?
2.	A pictorial depiction of the sources of bias would improve the clarity of Section 4. Alternatively, referring to Figure 2 could help.
3.	In Figure 2, it appears that the authors are suggesting the use of bias mitigation strategies in each step of the mode development process. If this is the case, a discussion of the deterioration of model performance that often accompanies such bias mitigation measures might be necessary.
4.	The spacing between the descriptions of each reference and related datasets in Table 2. For instance, for Soares et al. 2019, the Ford Motor dataset has a line break preceding it, making it seem like it’s associated with the next reference.

  a.	I also recommend dropping the reference text in the first column and the year column, since the citations already include the author names and year of publication.

**Strengths And Weaknesses:**

The paper addresses an oft-underappreciated aspect of AI development and a significant barrier to its widespread adoption in many fields – we need trustworthy and reliable AI systems for them to be more widely accepted. In that regard, this is a significant and welcome contribution as it provides a fairly comprehensive overview of the landscape of Trustworthy AI across many domains. Additionally, it appears that the authors have diverse backgrounds, which is essential for a rounded and informed discussion of Trustworthy AI, and its numerous potential applications. This is especially evident in Section 5, where they look at work on Trustworthy AI in a wide variety of applications and list an extensive array of related works in Table 2.

I would also like to laud the authors for their attempt to untangle the complexities of this space and define a series of principles and concepts underpinning Trustworthy AI, as can be seen in Figure 1. However, this effort falls short in that a lot of these concepts seem to overlap significantly, making it difficult to appreciate the differences between them; one would expect fundamental concepts to be distinct pillars underpinning the wider discussion on Trustworthy and Responsible AI. For instance, the concepts of Fairness and Responsible AI, Explainable AI and Communication, AI Governance and Regulatory Compliance – each of these pairs overlaps significantly and it is not apparent if or why they should be considered separately when thinking of Trustworthy AI. This confusion manifests itself in Sections 2 and 3, where the discussion oft comes across as highly repetitive. Brevity should be the goal here for the sake of both clarity and readability; perhaps a more hierarchical outline of these foundation pillars would be more appropriate. For instance, fairness could be discussed alongside responsible AI or as a concept inherent to the discussion of responsible AI.

---

> ### Author Response · Authors · 2025-08-03
> **Response to Reviewer AwCm’s Comments**
>
> We sincerely appreciate the time and careful consideration you have given to reviewing our manuscript. We have addressed all your valuable comments and suggestions in the revised version, with changes highlighted in yellow for easy reference. Each modification was carefully considered to improve the quality and clarity of our work. Please find our point-by-point responses below, and we would be happy to provide any additional clarifications if needed.
>
> **Comment:**
> Pursuant to the comments from the previous section, I would certainly like to see an effort made to amend the text in Sections 2 and 3 to improve the clarity of the text. This is important as these serve as key contributions of this work.
>
> a.	It is not clear what the distinction between explainability and interpretability is.
>
> **Response:**
> We sincerely appreciate your constructive feedback. In response to your comment, we have revised Sections 2 and 3 to improve clarity, particularly regarding the distinction between explainability and interpretability.
>
> **Comment:**
> On Page 3, the paper names anonymization and federated learning as potential solutions to patient privacy. Both these strategies have to be shown to be ineffective in adversarial scenarios. A qualification of where these strategies can be effective is needed here. A similar claim is made in the first paragraph of Section 3.
>
> **Response:**
> We thank the reviewer for the insightful comment. We have incorporated the suggested qualification in both locations, clarifying the limitations of anonymization and federated learning in adversarial scenarios.
>
> **Comment:**
> The mathematical definition of demographic parity in Table 1 appears to be incorrect.
>
> **Response:**
> Thank you for catching this oversight. We have corrected the mathematical definition of demographic parity in Table 1.
>
> **Comment:**
> When discussing autonomy on Page 2, it is stated that “In autonomy, the aim is to trade-off decisions by human against decisions made my machine”. How is this autonomy?
>
> **Response:**
> We apologize if our original explanation of autonomy was unclear. We have now revised the discussion on Page 3 to provide a clearer definition of autonomy in this context.
>
> **Comment:**
> In Figure 2, it appears that the authors are suggesting the use of bias mitigation strategies in each step of the mode development process. If this is the case, a discussion of the deterioration of model performance that often accompanies such bias mitigation measures might be necessary.
>
> **Response:**
> Thank you for this valuable observation. We have now added a discussion in Section 4.4 (Page 16) addressing the potential trade-offs between bias mitigation and model performance.
>
> **Comment:**
> The spacing between the descriptions of each reference and related datasets in Table 2. For instance, for Soares et al. 2019, the Ford Motor dataset has a line break preceding it, making it seem like it’s associated with the next reference.
>
> a. I also recommend dropping the reference text in the first column and the year column, since the citations already include the author names and year of publication.
>
> **Response:**
> We sincerely appreciate your careful review of Table 2. We have implemented both of your suggested improvements.

---

> > ### Comment · Reviewer_AwCm · 2025-08-06
> > **Some concerns not sufficiently addressed**
> >
> > I thank the authors for their effort in addressing the reviewer concerns. Particularly, the paper is now much clearer in its delineation of the various principles.
> >
> > I believe my concerns have been addressed with these edits. As for concerns raised by the other reviewers, the only remaining concern is with the “genericness” of the paper. But as that reviewer appears to be satisfied with these changes, I am happy to recommend that this paper be accepted.

---

> > > ### Comment · Reviewer_AwCm · 2025-08-06
> > > **Incorrect title in previous comment**
> > >
> > > Sorry, the title of my previous comment is incorrect. Please ignore that for the purposes of this discussion.

---

> > > > ### Author Response · Authors · 2025-08-06
> > > > **Response to Reviewer AwCm’s Comment**
> > > >
> > > > We sincerely appreciate your comments and thank you for recommending our manuscript for acceptance.

---

### Comment · Action_Editor_quqG · 2025-07-29
**Discussion**

Dear reviewers, dear authors,

This submission has now received all 3 reviews. The authors have already replied to the first two reviews.

@Reviewers: Please engage with the replies from the authors and continue the discussion.
@Authors: Please note that the third review has been posted over a week ago and you are welcome to reply to it.

Given the advanced timeline, I will leave the discussion open for a week from now, before I will ask reviewers to submit an official recommendation.

Best,
AE

---

### Decision · Action_Editor_quqG · 2025-08-07

**Recommendation:** Accept as is

**Audience:**

Yes

**Audience Explanation:**

Despite its broad scope, the paper addresses widely relevant themes in trustworthy and responsible AI, which are of ongoing interest to the TMLR audience.

**Claims And Evidence:**

Yes

**Claims Explanation:**

The reviewers agreed that the claims made in the submission are supported by accurate and clear evidence. The authors responded thoroughly to feedback, clarified key concepts, corrected technical inaccuracies, and improved clarity, leading all reviewers to endorse the paper’s acceptance despite some concerns about breadth and novelty, which are explicitly not reasons to reject at TMLR.